EMBO
*reports*

*Report*

# PPM1G forms a PPP-type phosphatase holoenzyme with B56δ that maintains adherens junction integrity

Parveen Kumar[1,2], Prajakta Tathe[1,2], Neelam Chaudhary[1] & Subbareddy Maddika[1,*]

## Abstract

Serine/threonine phosphatases achieve substrate diversity by forming distinct holoenzyme complexes in cells. Although the PPP family of serine/threonine phosphatase family members such as PP1 and PP2A are well known to assemble and function as holoenzymes, none of the PPM family members were so far shown to act as holoenzymes. Here, we provide evidence that PPM1G, a member of PPM family of serine/threonine phosphatases, forms a distinct holoenzyme complex with the PP2A regulatory subunit B56δ. B56δ promotes the re-localization of PPM1G to the cytoplasm where the phosphatase can access a discrete set of substrates. Further, we unveil α-catenin, a component of adherens junction, as a new substrate for the PPM1G-B56 phosphatase complex in the cytoplasm. B56δ-PPM1G dephosphorylates α-catenin at serine 641, which is necessary for the appropriate assembly of adherens junctions and the prevention of aberrant cell migration. Collectively, we reveal a new holoenzyme with PPM1G-B56δ as integral components, in which the regulatory subunit provides accessibility to distinct substrates for the phosphatase by defining its cellular localization.

**Keywords** adherens junction; B56δ; PP2A; PPM1G; α-catenin
**Subject Categories** Cell Adhesion, Polarity & Cytoskeleton; Post-translational Modifications & Proteolysis; Signal Transduction

## Introduction

Protein phosphorylation is a ubiquitous reversible post-translation modification involved in regulation of several biological processes [1,2]. In humans, 98% of total phosphorylation occur on serine/threonine residues and is regulated by counter actions of protein kinases and phosphatases [3,4]. Ser/Thr residues are dephosphorylated by members belonging to three different phosphatase families: (i) PPPs—phosphoprotein phosphatases, (ii) PPMs—metal ($Mg^{2+}$/$Mn^{2+}$)-dependent protein phosphatases, and (iii) FCP/SCPs—aspartate-based TFIIF-associating component of RNA polymerase II CTD phosphatase/small CTD phosphatases [5,6]. While PPP and PPM phosphatase act on various cellular substrates, members of FCP/SCP family have only one substrate—the CTD of RNA polymerase II [7–9]. PPP phosphatases are subdivided into seven groups (PPP1-PPP7). Among PPP family, PPP2/PP2A contributes to majority of phosphatase activity inside the cell. PP2A assembles into nearly 100 different heterotrimeric holoenzymes, each containing a common core enzyme, a stable heterodimer of the scaffold (A) subunit and catalytic (C) subunit, and a variable mutually exclusive regulatory (B) subunit [10,11]. It is due to this unique ability to form holoenzymes, PP2A can dephosphorylate numerous targets specifically and participate in diverse cellular processes. Regulatory subunits in holoenzyme are required to control localization, activity, and/or substrate specificity of the phosphatase [12]. In humans, PP2A regulatory subunits occur in four distinct gene families, B (B55), B′ (B56), B″ (PR48/PR72/PR130), and B‴ (PR93/PR110/striatin), each group contains two to five isoforms. Although many regulatory subunits of PPP phosphatases have been identified, their functions and specific substrate information are still limited.

On the other hand, PPM phosphatase family consists of at least 17 members, which differ in protein sequence to PPP family but employ a similar catalytic mechanism. PPM phosphatases are dependent on $Mg^{2+}$ or $Mn^{2+}$ ions for their activity and are insensitive to phosphatase inhibitors okadaic acid and microcystin [13]. Importantly, PPM phosphatases are so far shown to function exclusively as monomers and they utilize non-catalytic domains in their primary structure for their substrate recognition and subcellular localization. Till date, no regulatory subunits have been found for PPM phosphatases, and therefore, no holoenzymes for this family of phosphatases were reported.

Here, in the process of mapping new functions for PP2A regulatory subunits, we surprisingly found that one of the PP2A regulatory subunits (B56δ) forms a holoenzyme complex with PPM1G, a member of PPM phosphatase family. We found that the regulatory subunit helps in diversifying the substrate availability for PPM1G phosphatase by regulating its subcellular localization. In this case, we identified α-catenin, a cell junction protein, as a novel substrate

---

1   Laboratory of Cell Death & Cell Survival, Centre for DNA Fingerprinting and Diagnostics (CDFD), Uppal, Hyderabad, India
2   Graduate Studies, Manipal Academy of Higher Education, Manipal, India
    *Corresponding author. Tel: +91 40 2721 6168; E-mail: msreddy@cdfd.org.in

for B56δ-PPM1G holoenzyme. Our data demonstrate that B56δ-PPM1G is required for proper assembly of adherens junction via dephosphorylating α-catenin.

## Results and Discussion

### Identification of PPM1G-B56δ phosphatase holoenzyme complex

We utilized an interaction proteomic approach and isolated protein complexes associated with PP2A regulatory subunits to identify new cellular functions of PP2A complexes. Nine regulatory subunits of PP2A (Table EV1) were individually cloned into a triple-epitope (S-protein, Flag, and streptavidin-binding peptide)-tagged (SFB) construct. Tandem affinity purification with streptavidin–agarose beads and S-protein–agarose beads using lysates derived from cells expressing these tagged proteins followed by mass spectrometry analysis allowed us to discover protein complexes associated with these regulatory subunits. By using a SAINT score cutoff of 0.8, we identified 297 high confident interactions (HCIs) associated with nine PP2A regulatory subunits (Dataset EV1), among which, several known as well as unknown interactions were mapped. Unexpectedly, we found that one of the PPP regulatory subunit PPP2R5D (or B56δ) to be associated with PPM1G, a member of PPM family of serine/threonine phosphatases (Fig 1A). PPM1G also known as PP2Cγ is a $Mg^{2+}/Mn^{2+}$-dependent nuclear serine/threonine phosphatase that belongs to PPM phosphatase family [14]. It plays an important role in different functions such as nucleosome assembly, protein translation, apoptosis, mRNA splicing, and DNA damage response [15–19].

We confirmed the interaction of endogenous PPM1G specifically with B56δ in cells (Fig 1B). Further, bacterially purified recombinant GST-tagged B56δ and MBP-tagged PPM1G proteins interacted with each other, suggesting a direct interaction between B56δ and PPM1G (Fig EV1A). As B56δ is a bonafide regulatory subunit of PP2A phosphatase, next we tested whether PPM1G is part of the PP2A holoenzyme complex. Interestingly, while PPP2R1A-B56δ interacts with each other in vitro, B56δ but not PPP2R1A specifically

interacts with PPM1G (Fig 1C). Further, though B56δ interacted strongly with PPM1G in cells, neither of the catalytic subunits (PPP2CA and PPP2CB) nor of the scaffold structural subunits (PPP2R1A and PPP2R1B) of PP2A interacted with PPM1G (Fig 1D), suggesting that PPM1G-B56δ is a distinct holoenzyme formed independent of PPP holoenzyme complex. All the B56 family members (B56α, B56β, B56γ, B56δ, B56ε) have a common central B56 domain with high protein sequence homology (known to bind PP2A enzyme), flanked with variable N- and C-terminal regions (Appendix Fig S1). Moreover, we found a small fraction of B56β and B56ε in B56δ interaction list (Dataset EV1). Therefore, we tested whether PPM1G associates with other B56 members as well. B56β weakly associates with B56δ (Fig EV1B); however, PPM1G was found to specifically interact with only B56δ subunit but not with other members of this family (Fig 1E). To further understand how B56δ specifically interacts with PPM1G, we sought to identify the PPM1G-interacting region on B56δ by using various deletion mutants (Fig EV1C). N-terminal region of B56δ was found to be essential for its interaction with PPM1G (Fig EV1D). N-terminus is variable among different B56 family members, and therefore, PPM1G could specifically interact with B56δ but not others. Further fine mapping has revealed that a region of amino acids between 80 and 102 of B56δ is required for interaction with PPM1G (Fig EV1E). On the other hand, B56 subunit is known to interact with PP2A holoenzyme complex via its B56 domain [10,11]. Based on the contact points between B56 and PPP2R1A subunits, derived from the PP2A holoenzyme crystal structure, we generated various point mutants of B56 region of B56δ. Our interaction analysis with these mutants has suggested that conserved residues at L183, H282, Y289, and E290 positions are critical for interaction between B56δ and PPP2R1A (Fig EV1F). Importantly, deletion of a region between amino acids 91 and 102 (Fig 1F) severely reduced B56δ binding to PPM1G in both cells and in vitro (Figs 1G and EV1G) with no effect on B56δ-PPP2R1A interaction (Figs 1H and EV1H). However, L183A or H282A mutants that show defective binding to PPP2R1A did not affect the interaction between B56δ and PPM1G, thus suggesting B56δ interacts with two independent phosphatase holoenzyme complexes via two distinct non-overlapping regions. Next, we

▶

**Figure 1. PPM1G-B56δ is a new phosphatase holoenzyme complex.**

A   Interaction network of B56δ with its associated proteins identified in this study. Blue line indicates B56δ interaction with PPM1G. Red dashed line represents interaction with PP2A complex proteins.

B   HEK293T cells were transfected with SFB control, SFB-tagged PR130, and SFB-B56δ, and the cell lysates were pulled down using streptavidin beads, and the interaction of endogenous PPM1G was detected by Western blot (WB).

C   Glutathione Sepharose beads bound with bacterially expressed recombinant GST, GST-B56δ, or GST-PPP2R1A proteins were incubated with bacterially purified recombinant MBP-PPM1G, and interaction of B56δ with PPM1G was detected by immunoblotting with MBP antibody. Interaction of B56δ-PPP2R1A was shown similarly using MBP-B56δ and GST-PPP2R1A proteins. Recombinant protein expression is shown by Coomassie staining.

D   HEK293T cells were transfected with SFB control, SFB-PPP2CA, SFB-PPP2CB, SFB-PPP2R1A, SFB-PPP2R1B, and SFB-B56δ, and the lysates were pulled down using streptavidin beads, and the interaction of endogenous PPM1G was detected by Western blot using PPM1G antibody.

E   Cell lysates from HEK293T cells expressing SFB vector, SFB-B56α, SFB-B56β, SFB-B56γ, SFB-B56δ, and SFB-B56ε were pulled down, and interaction with PPM1G was detected by Western blotting using PPM1G antibody.

F   Schematic representation of B56δ full length (FL), along with its various deletion and point mutants.

G   Cells expressing B56δ full length (FL), Δ91-102, L183A, and H282A mutants were pulled down with streptavidin beads, and interaction with PPM1G was detected by Western blotting.

H   Various mutants of B56δ were transfected along with myc-PPP2R1A and pulled down using streptavidin beads, and interaction with myc-PPP2R1A was detected by immunoblotting.

I   Binding affinities of the B56δ with PPM1G and PPP2R1A subunit were determined by bio-layer interferometry. 6xHis-tagged PPM1G and PPP2R1A were immobilized on Ni-NTA biosensors and incubated with the GST-B56δ wild type (WT) or Δ91–102 mutant at various concentrations (100–400 nm). Curves represent experimental trace obtained from the BLI experiments. Binding affinities ($K_d$) (± SD; $n$ = 3 independent experiments) for B56δ-PPM1G and B56δ-PPP2R1A are shown.

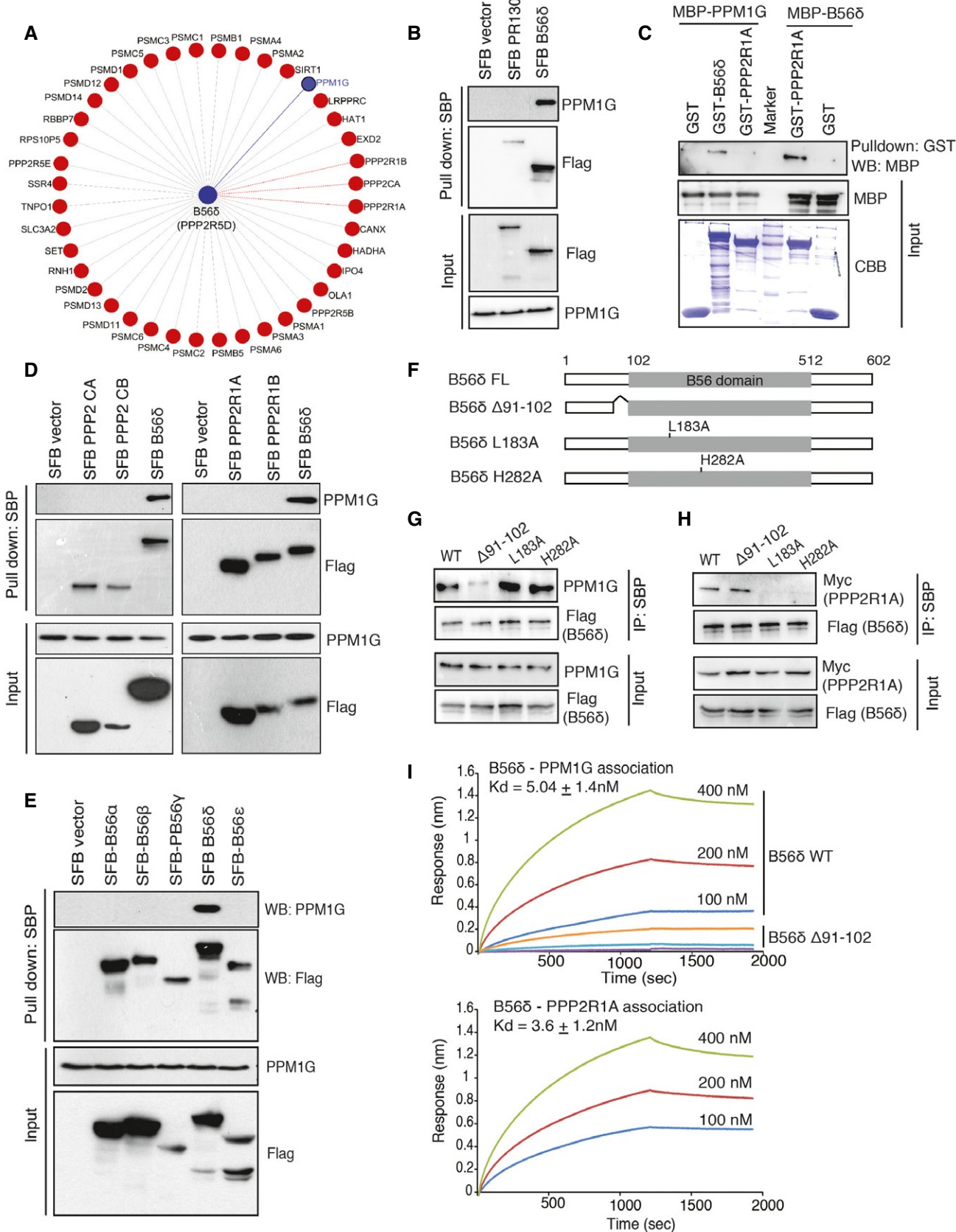

**Figure 1.**

measured the relative binding kinetics of B56δ interaction with two distinct holoenzyme complexes. We found that wild-type B56δ, but not a mutant lacking amino acids 91–102 (Δ91–102), binds with PPM1G with a $K_d$ of 5.04 ± 1.4 nM (Fig 1I). B56δ L183A and H282A mutants that are defective in binding with PPP2R1A show similar binding kinetics (5.59 ± 3.1 nM and 4.38 ± 2.1 nM, respectively) as wild-type B56δ with PPM1G (Fig EV1I), again supporting the ability of B56δ to form two independent holoenzyme complexes. Intriguingly, B56δ-PPM1G displays very close binding kinetics as of B56δ-PPP2R1A interaction ($K_d$ = 3.6 ± 1.2 nM). With closely similar binding kinetics, next we tested whether PPM1G forms a complex with B56δ by displacing PPP2R1A subunit from the B56δ-PP2A holoenzyme complex. However, *in vitro* competition experiments with excess of recombinant PPM1G did not affect the binding of PPP2R1A with B56δ (Fig EV1J), thus again supporting the assembly of a novel B56δ-PPM1G complex distinct from the known B56δ-PP2A holoenzyme complex.

## B56δ controls cellular localization of PPM1G

Regulatory subunits are critical components in the PP2A holoenzyme complex, wherein they largely control the substrate specificity. Additionally, regulatory subunits are known to function in holoenzyme by controlling either specific activity of the phosphatase or spatial distribution of the holoenzyme to different cellular compartments [20]. We therefore tested whether the regulatory subunit B56δ alters the cellular localization of PPM1G. In our immunofluorescence experiments while PPM1G was predominantly localized in the nucleus with sparse cytoplasmic content, upon overexpression of B56δ, a significant portion of PPM1G was redistributed to the cytosol in two independent breast cancer cell lines (Fig 2A and B). Biochemical subcellular fractionation experiments also confirmed that expression of B56δ (Fig 2C) led to cytoplasmic accumulation of PPM1G (Fig 2D and Appendix Fig S2A). Effect of B56δ on PPM1G cytoplasmic translocation is specific as none of the other tested regulatory subunits affected PPM1G localization (Fig EV2A). On the other hand, depletion of B56δ in cells prevents cytoplasmic localization of PPM1G (Fig 2E and Appendix Fig S2B). It is possible that B56δ promotes cytoplasmic accumulation of PPM1G through either promoting the nuclear export of PPM1G or sequestering the cytoplasmic fraction of PPM1G. We tested these possibilities by performing localization experiments in the presence of Leptomycin B (LMB), a potent and specific nuclear export

inhibitor. Treatment of cells with LMB did not prevent the cytoplasmic accumulation of PPM1G caused by B56δ (Fig EV2B), suggesting that B56δ controls PPM1G localization by cytoplasmic retention and not through nuclear export. Furthermore, we tested whether B56δ-PPM1G interaction is required for PPM1G presence in the cytoplasm. While expression of full-length B56δ promotes cytoplasmic PPM1G localization, mutant B56δ that lacks PPM1G binding (Δ91–102) was unable to retain PPM1G in cytoplasm (Fig 2F and G). In contrast, B56δ mutants defective in binding to PP2A (L183A, H282A) still maintain their ability to drive cytoplasmic localization of PPM1G, similar to wild-type B56δ. Together, these data fully support our hypothesis that B56δ controls PPM1G localization in cells by forming a specific complex independent of PP2A holoenzyme.

## α-Catenin is a novel substrate of PPM1G-B56δ phosphatase holoenzyme

We next sought to identify the functional substrates of the newly identified PPM1G holoenzyme. Tandem affinity purification using cell lysates expressing a triple-epitope (S-protein, Flag, and streptavidin-binding peptide)-tagged version of PPM1G followed by mass spectrometric identification revealed several known as well as unknown PPM1G-associated proteins (Fig EV3A). Among several interacting proteins, we found α-catenin (CTNNA1), an intercellular adhesion protein and a critical component of adherens junction [21,22] to be a novel PPM1G-associated protein. By performing immunoprecipitation with α-catenin antibody, we confirmed the endogenous interaction of PPM1G with α-catenin in cells (Fig 3A). The existence of PPM1G-α-catenin complex in cells was further demonstrated by interaction of exogenously expressed proteins in 293T cells (Fig EV3B). Although PPM1G was denoted as a nuclear phosphatase, we demonstrated that a significant amount of protein is also found in cytoplasm, which is driven by B56δ. Our immunoprecipitation analysis after cellular fractionation suggested that while known interaction of α-catenin and β-catenin occurs in cytoplasm as well as in the nucleus, in contrast only cytoplasmic PPM1G specifically interacts with α-catenin (Fig EV3C). Further, interaction of PPM1G-α-catenin was significantly reduced in B56δ-depleted cells (Fig 3B and Appendix Fig S3A), thus supporting that B56δ-driven cytoplasmic localization of PPM1G is essential for its interaction with new substrate in the cytoplasm.

**Figure 2. B56δ retains PPM1G in cytoplasm.**

A  MCF7 and MDA-MB-231 cells transfected with MYC-B56δ were fixed and imaged using a confocal microscope after staining with antibodies against MYC tag and PPM1G to determine PPM1G localization. Representative images from three independent experiments are shown. Scale bars, 5 μm.

B  Quantification of cytoplasmic-to-nuclear (Cyto/Nuc) ratio of PPM1G intensity per each cell was plotted ($n$ = 35 for MCF-7 cells; $n$ = 25 for MDA-MB-231 cells). Error bars indicate SD. ***$P$ < 0.001 (one-way ANOVA, post hoc test: Bonferroni's multiple comparison test).

C  Western blot showing expression levels of SFB-tagged B56δ along with endogenous B56δ using indicated antibodies.

D  Cells expressing either vector or SFB-B56δ were fractionated into nuclear and cytoplasmic fractions, and levels of indicated proteins were determined using Western blotting. Tubulin and histone H3 were used to mark cytoplasmic and nuclear fractions, respectively.

E  Cells treated with control or B56δ shRNA were fractionated into cytoplasmic and nuclear fractions, and levels of indicated proteins were determined using immunoblotting. Tubulin and histone H3 were used to mark cytoplasmic and nuclear fractions, respectively.

F  Localization of PPM1G in cells expressing either empty vector (control), full-length B56δ (FL), or various indicated mutants was determined by confocal imaging after staining with Flag and Myc antibodies. Representative images from two independent experiments are shown. Scale bars, 10 μm.

G  Cytoplasmic-to-nuclear (Cyto/Nuc) ratio of PPM1G intensity per each cell expressing either vector, B56δ full length (FL), or mutant versions was quantified and plotted ($n$ = 10). Error bars represent SD. ***$P$ < 0.001 (one-way ANOVA, post hoc test: Bonferroni's multiple comparison test).

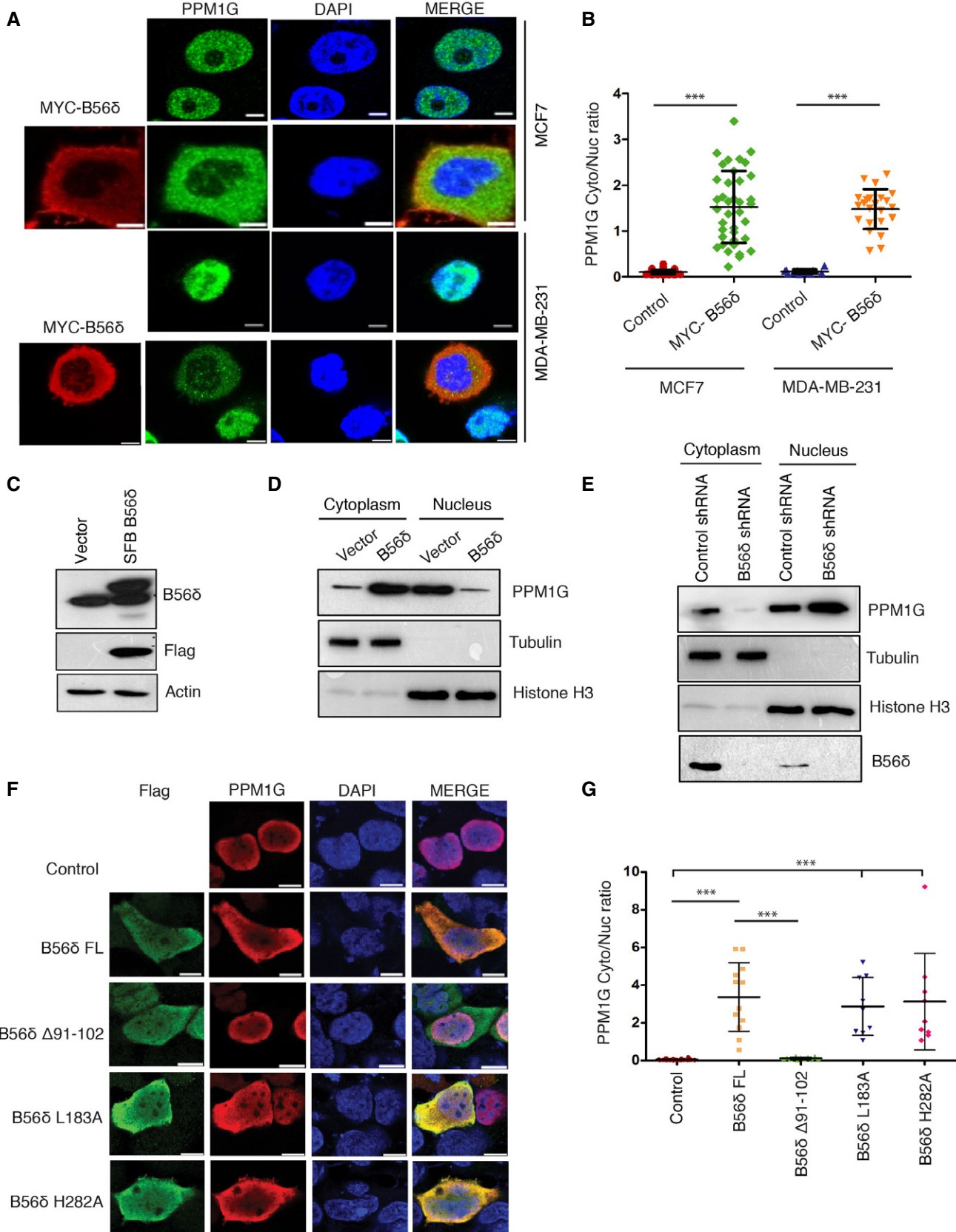

**Figure 2.**

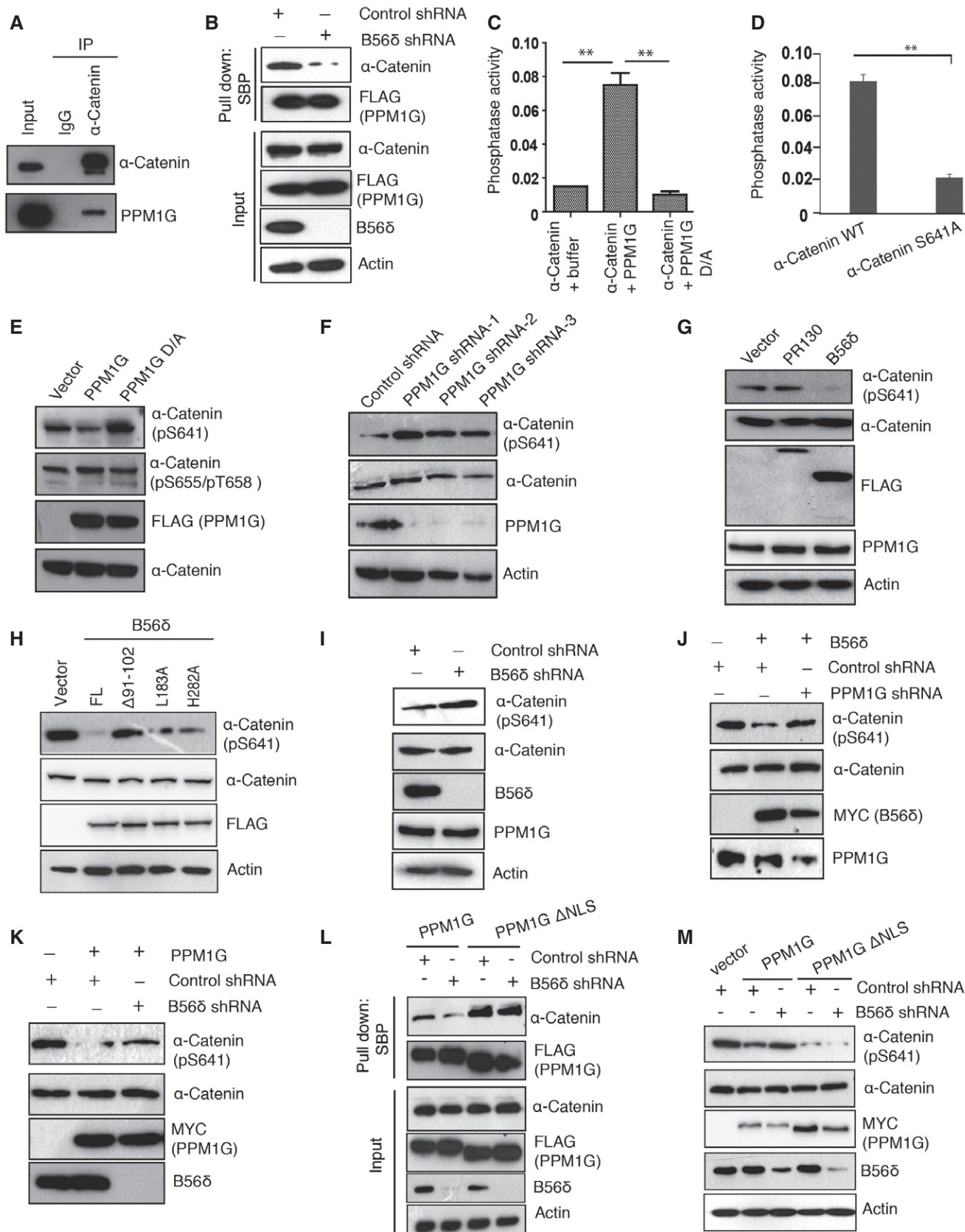

Figure 3.

◀

**Figure 3. α-Catenin is a new cytoplasmic substrate of PPM1G-B56δ holoenzyme.**

A   Endogenous association of PPM1G with α-catenin was detected by immunoblotting with PPM1G antibody after performing immunoprecipitation with either IgG or α-catenin antibody.

B   SFB-PPM1G was transfected in both control shRNA and B56δ shRNA cells and pulled down using streptavidin beads. Interaction with α-catenin was determined by Western blotting.

C   *In vitro* phosphorylated MBP-tagged α-catenin was incubated with equal amounts of bacterially purified recombinant wild type and catalytically inactive mutant (D496A) of PPM1G, and the amount of released phosphate was assayed colorimetrically using the malachite green reagent (A620 nm). Data represent mean absorbance from three independent experiments. Error bars indicate SD, **P < 0.01 (one-way ANOVA, post hoc test: Bonferroni's multiple comparison test).

D   *In vitro* phosphate release assay was performed using wild-type α-catenin and S641A mutant as substrates, and PPM1G activity on these proteins is plotted, n = 3. Error bars indicate SD; **P < 0.01 (one-way ANOVA, post hoc test: Bonferroni's multiple comparison test).

E   MDA-MB-231 breast cancer cells were transfected with vector, wild-type PPM1G, or catalytically inactive PPM1G D496A (D/A) mutant. Levels of phosphorylated α-catenin were detected by using phospho-specific antibodies.

F   Cells treated with control shRNA and PPM1G-specific shRNAs were lysed and blotted against specific antibodies.

G   Cells transfected with vector, SFB-PR130, and SFB-B56δ were lysed, and levels of pS641 α-catenin were detected using Western blotting.

H   293T cells transfected with either empty vector, full-length B56δ, or indicated mutants were lysed, and the levels of phosphorylated α-catenin were determined using pS641 α-catenin specific antibody.

I   Cells treated with control shRNA and B56δ-specific shRNAs were lysed, and the levels of phosphorylated α-catenin were determined using pS641 α-catenin specific antibody.

J, K   Cells were transduced with shRNAs as indicated, and the effect of (J) B56δ or (K) PPM1G expression on α-catenin phosphorylation was assessed.

L   SFB-PPM1G and SFB-PPM1G ΔNLS were transfected in both control shRNA and B56δ shRNA-transduced cells. Lysates were pulled down using streptavidin beads, and interaction with α-catenin was detected by Western blotting using α-catenin antibody.

M   Control shRNA and B56δ shRNA cells were transfected with vector, MYC-PPM1G, and MYC-PPM1G ΔNLS in indicated combination, and pS641 α-catenin levels were determined using Western blotting.

In order to understand the role of PPM1G and α-catenin interaction, we mapped the minimum region of PPM1G required for their interaction. Several PPM1G deletions were generated (Fig EV3D), and our co-immunoprecipitation results suggested that α-catenin binds to acidic-rich region of PP1MG (Fig EV3E). The acidic-rich region was designated as a substrate-binding domain of PPM1G [23]. Thus, we next tested whether α-catenin is a substrate of PPM1G. We found that active PPM1G, but not a catalytically dead mutant, readily dephosphorylates α-catenin (Fig 3C). Earlier studies have shown that CK2α phosphorylates α-catenin at S641 residue to release its binding from β-catenin [24]. We next tested whether this residue is dephosphorylated by PPM1G. While PPM1G could release phosphate from wild-type α-catenin *in vitro*, it fails to do so with S641A mutant (Fig 3D). Further, by using a phospho-specific antibody, we found that phosphorylation at S641 residue is reduced in MDA-MB-231 cells stably expressing wild-type PPM1G but not catalytically inactive D496A mutant (Fig 3E and Appendix Fig S3B). On the other hand, depletion of PPM1G enhanced phosphorylation of S641 residue (Fig 3F and Appendix Fig S3C). Therefore, S641 is a PPM1G dephosphorylation site on α-catenin.

Similar to PPM1G, overexpression of B56δ led to reduced phosphorylation of α-catenin at S641 residue (Fig 3G and Appendix Fig S3D). Notably, expression of wild-type B56δ or PP2A binding defective mutants (L183A and H282A), but not PPM1G binding defective mutant (Δ91–102), reduced α-catenin phosphorylation (Fig 3H and Appendix Fig S3E). Conversely, siRNA-mediated depletion of B56δ in cells enhanced α-catenin phosphorylation (Fig 3I and Appendix Fig S3F), suggesting that B56δ is necessary for α-catenin dephosphorylation. Given that both PPM1G and B56δ suppressed α-catenin phosphorylation, we next tested whether the holoenzyme assembly of PPM1G-B56δ is required for α-catenin dephosphorylation. Overexpression of B56δ in PPM1G-depleted cells could not efficiently dephosphorylate α-catenin (Fig 3J and Appendix Fig S3G). Similarly, expression of PPM1G in B56δ-depleted conditions is not sufficient to remove α-catenin

phosphorylation (Fig 3K and Appendix Fig S3H). These results suggest that PPM1G and B56δ are interdependent on each other and function cooperatively as a holoenzyme wherein PPM1G acts as a catalytic phosphatase unit and B56δ controls appropriate localization of the phosphatase.

In the holoenzyme, if B56δ is critical for cytoplasmic localization of the PPM1G phosphatase, we hypothesized that forcible expression of PPM1G in cytoplasm may overcome the dependence of PPM1G on B56δ. Deletion of NLS region that localizes PPM1G exclusively in the cytoplasm (Fig EV3F) is sufficient to restore the interaction of PPM1G and α-catenin that was lost upon B56δ depletion in cells (Fig 3L and Appendix Fig S3I). Concomitantly, unlike PPM1G wild type that cannot dephosphorylate α-catenin in the absence of B56δ, PPM1G ΔNLS mutant readily dephosphorylates α-catenin irrespective of B56δ status in the cells (Fig 3M and Appendix Fig S3J). Thus, these data fully support our hypothesis that PPM1G-B56δ cooperatively functions as a holoenzyme to act on substrates in cytoplasm.

## PPM1G-B56δ holoenzyme is required for assembly of adherens junctions

α-catenin is an obligate component of adherens junctions where in concert with β-catenin it physically links the cytoplasmic domain of cadherin complex to the actin cytoskeleton [21,25]. α-catenin is indispensable for the structural maintenance and function of adherens junctions [26]. As PPM1G-B56δ holoenzyme modulated α-catenin phosphorylation, we examined whether this phosphatase complex is required for proper assembly of adherens junction. Indeed, while in control MCF-7 breast cancer cells α-catenin was appropriately localized to cell junctions at the membrane, depletion of either PPM1G or B56δ led to diminished localization of α-catenin to cell membrane (Fig 4A and B), suggesting that α-catenin dephosphorylation is essential for its localization to cell junctions. Similar observations were made in MCF10A breast epithelial cells (Fig EV4A). Further, in agreement with this fact, immunofluorescent

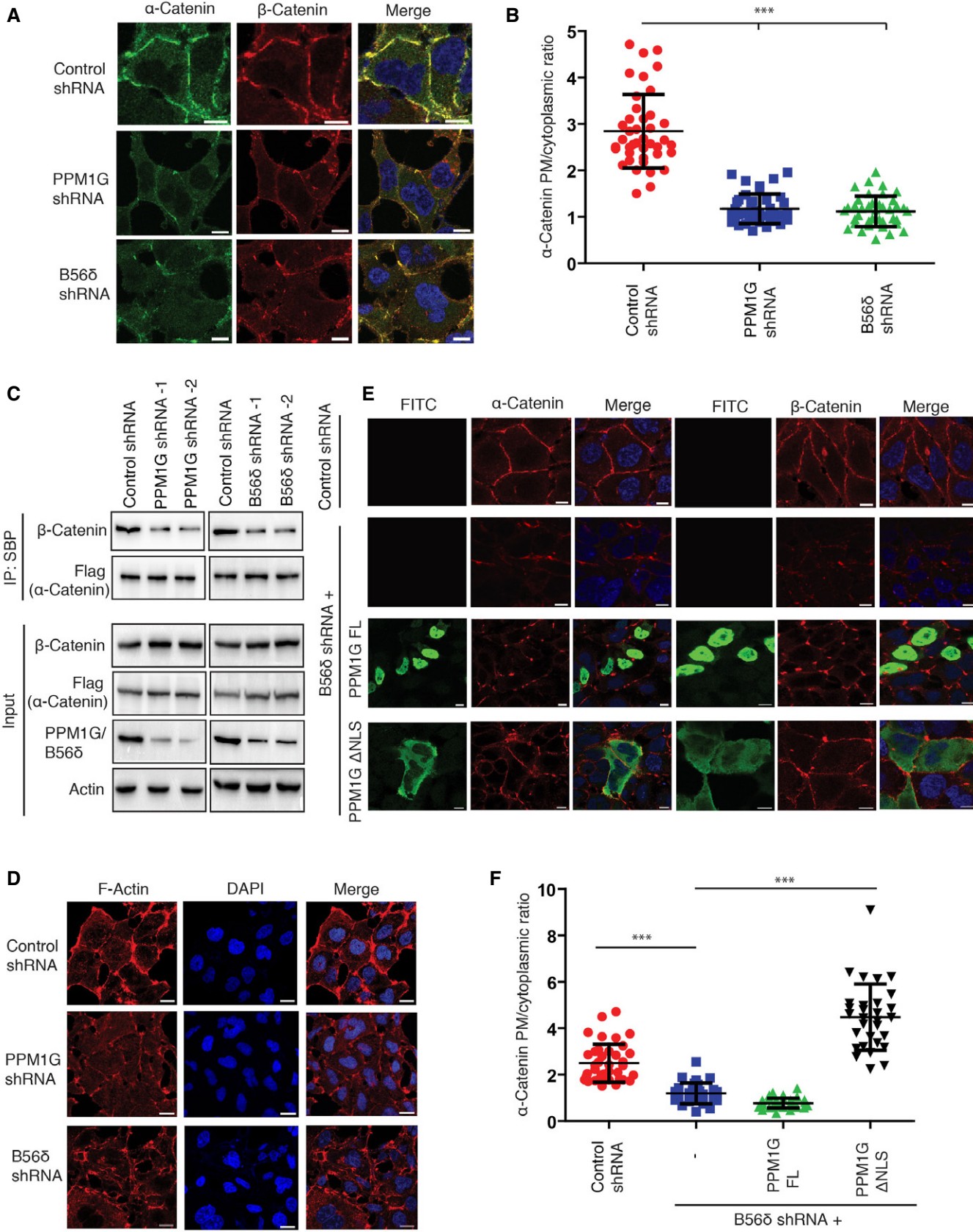

**Figure 4.**

**Figure 4. PPM1G-B56δ holoenzyme is required for adherens junction assembly.**

A  Cells transduced with the indicated shRNAs were fixed, and localization of α-catenin and β-catenin was determined by imaging with a confocal microscope after staining with antibodies against α-catenin and β-catenin. Representative images from three independent experiments are shown. Scale bars, 10 μm.

B  Quantification of plasma membrane (PM)-to-cytoplasmic ratio of α-catenin intensity per each cell from Fig 4A was plotted (n = 40 cells). Error bars indicate SD. ***P < 0.001 (one-way ANOVA, post hoc test: Bonferroni's multiple comparison test).

C  Control, PPM1G, and B56δ shRNA-transduced cells were transfected with SFB-α-catenin. Cell lysates were pulled down using streptavidin beads, and interaction with β-catenin was determined by Western blotting.

D  F-Actin in control and PPM1G-depleted cells was stained using rhodamine-labeled phalloidin, and the cytoskeleton arrangement was detected by using confocal microscope. Scale bars, 20 μm.

E  Localization of α-catenin and β-catenin in control shRNA cells or B56δ-depleted cells with or without expression of SFB-PPM1G and SFB-PPM1G ΔNLS was detected by imaging with confocal microscope after staining with specific antibodies. Representative images are shown. Scale bars, 10 μm.

F  Quantification of plasma membrane (PM)-to-cytoplasmic ratio of α-catenin intensity per each cell from (E) was plotted (n = 30 cells). Error bars indicate SD. ***P < 0.001 (one-way ANOVA, post hoc test: Bonferroni's multiple comparison test).

staining using antibodies against phosphorylated S641 residue confirmed that phosphorylated form of α-catenin is exclusively in cytoplasm and nucleus but is unable to localize to cell junctions (Fig EV4B). Also, while wild type and a phospho-dead S641A mutant of α-catenin localizes to cell membrane, the constitutive phospho-mimetic S641D mutant was defective in membrane localization (Fig EV4C). Defective localization of α-catenin in PPM1G- and B56δ-depleted cells also led to diminished β-catenin at cell membrane (Fig 4A) and thus leading to disruptive adherens junctions. In fact, the association of α-catenin with β-catenin is compromised in PPM1G- and B56δ-depleted cells compared with control cells (Fig 4C). No significant changes in the protein levels of α-catenin and β-catenin were observed with depletion of PPM1G (Fig EV4D), thus suggesting that disruptive adherens junctions are due to mislocalization but not the protein stability of the cell junction proteins upon loss of holoenzyme. Furthermore, following the disruptive cell junction assembly, linking of actin cytoskeleton to adherens junction was severely affected in these conditions as evidenced by disorganized F-actin assembly upon PPM1G and B56δ depletion in cells (Figs 4D and EV4E). Next, we tested whether the adherens junction integrity is dependent on holoenzyme assembly. Overexpression of wild-type PPM1G was not sufficient to restore the lost integrity of adherens junction caused due to B56δ depletion in cells, thus supporting the requirement of an intact holoenzyme. However, PPM1G ΔNLS

mutant that bypasses the requirement of B56δ could readily rescue the junction defects resulted upon B56δ loss in cells (Fig 4E and F). Together, these data confirm that PPM1G-B56δ holoenzyme is required for proper assembly of adherens junction in cells.

## PPM1G-B56δ holoenzyme control cell migration

Functionally, α-catenin along with β-catenin and cadherins at adherens junction is intricately involved in dynamic remodeling of adhesive contacts and thereby controls cell adhesion and migration [27]. Thus, we next tested whether PPM1G-B56δ holoenzyme complex suppresses cell migration by modulating α-catenin. We found that expression of PPM1G, similar to α-catenin, strongly suppressed cell migration in invasive breast epithelial cancer cells in a catalytic activity-dependent manner (Fig 5A and B). Conversely, depletion of PPM1G using siRNA in MCF10A cells led to increased cell migration (Fig EV5A). To further validate the importance of PPM1G association with α-catenin in tumor cell phenotypes, we screened for PPM1G mutations in human cancers that might be defective in binding to its substrates. While our search for somatic mutations in the COSMIC database revealed several missense and nonsense mutations in PPM1G, we specifically focused on mutations occurring in acidic domain, where α-catenin binds. Interestingly, several of these point mutants (such as D133V, S173C, G188W, A237D, and A252S)

**Figure 5. PPM1G-B56δ complex controls cell migration by dephosphorylating α-catenin.**

A, B  MDA-MB-231 cells were transfected with vector control, PPM1G wild type, catalytically inactive PPM1G D496A mutant, or α-catenin, and transwell migration assay was performed. Phase-contrast microscopic images (A) were taken after staining the migrated cells with crystal violet, and (B) quantification data for migrated cells per field (derived from average of 3 different fields) from three independent experiments are shown. Error bars indicate standard deviation (n = 3), ***P < 0.001 (one-way ANOVA, post hoc test: Bonferroni's multiple comparison test).

C, D  MCF-7 cells were transfected with SFB vector, SFB-B56δ, or indicated mutants, and transwell migration assay was performed. Phase-contrast microscopic images (C) were taken after staining the migrated cells with crystal violet, and (D) quantification data for migrated cells per field (derived from average of 3 different fields) from three independent experiments are shown. Error bars indicate standard deviation (n = 3), ***P < 0.001 (one-way ANOVA, post hoc test: Bonferroni's multiple comparison test).

E, F  SFB-B56δ was expressed in control shRNA and PPM1G shRNA-transduced MCF7 cells. Representative images from transwell migration assay (E) are shown, and (F) quantified data for migrated cells per field from three independent experiments were plotted. Error bars indicate standard deviation (n = 3), ***P < 0.001 (one-way ANOVA, post hoc test: Bonferroni's multiple comparison test).

G, H  SFB-PPM1G was expressed in control shRNA and B56δ shRNA-transduced MCF7 cells. Representative images from transwell migration assay (G) are shown, and (H) quantified data for migrated cells per field from three independent experiments were plotted. Error bars indicate standard deviation (n = 3), ***P < 0.001 (one-way ANOVA, post hoc test: Bonferroni's multiple comparison test).

I, J  Control shRNA and B56δ shRNA-transduced MCF7 cells were transfected with MYC-PPM1G and MYC-PPM1G ΔNLS. Representative images from transwell migration assay (I) performed with these cells are shown, and (J) quantified data for migrated cells from three independent experiments per field were plotted. Error bars indicate standard deviation (n = 3), ***P < 0.001 (one-way ANOVA, post hoc test: Bonferroni's multiple comparison test).

K, L  SFB-tagged α-catenin S641A and S641D mutants were expressed in PPM1G shRNA and B56δ shRNA-transduced MCF7 cells as shown. Representative images from transwell migration assay (K) performed with these cells are shown, and (L) quantified data for migrated cells per field from three independent experiments were plotted. Error bars indicate standard deviation (n = 3), ***P < 0.001 (one-way ANOVA, post hoc test: Bonferroni's multiple comparison test).

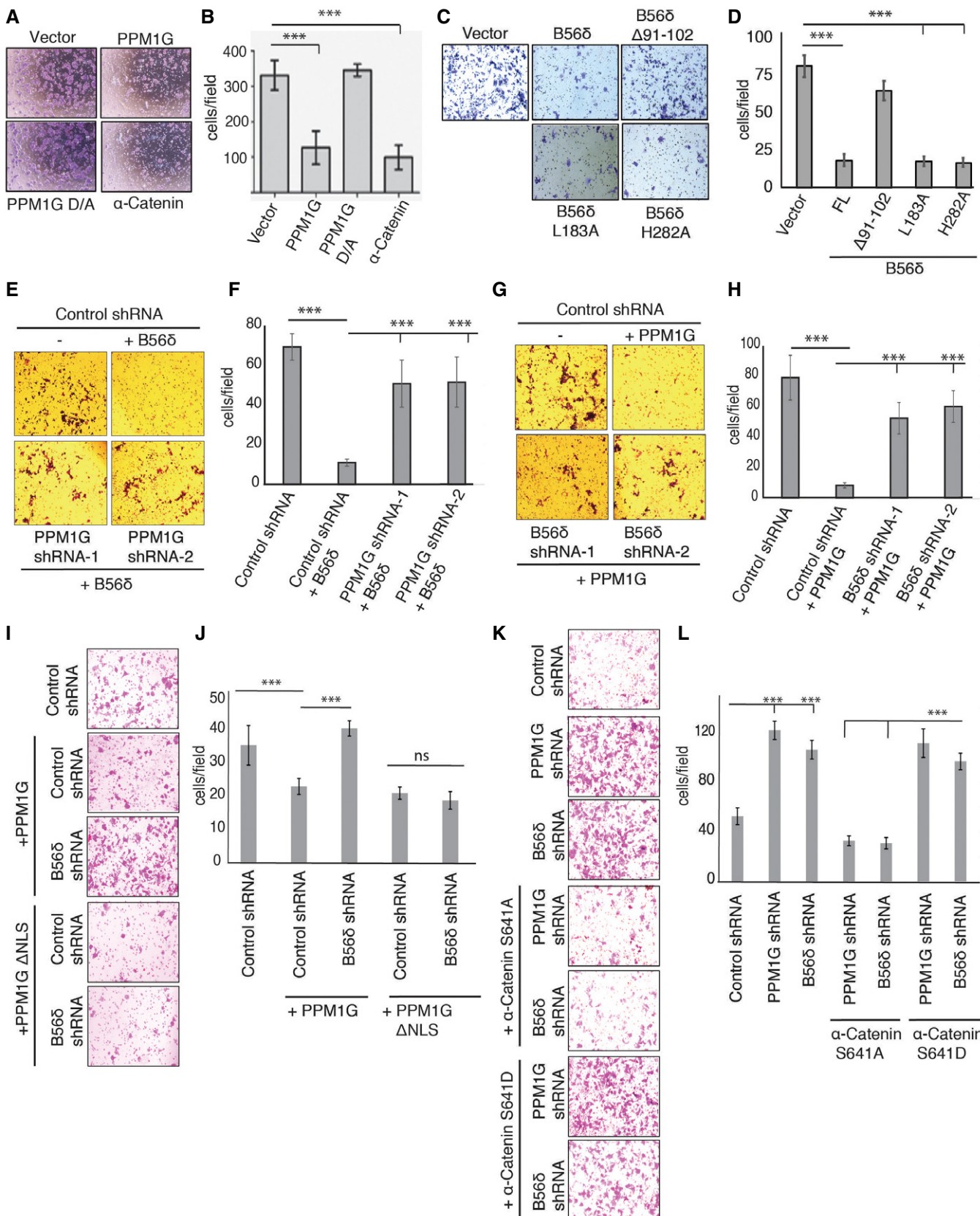

Figure 5.

show reduced binding to α-catenin compared with wild-type PPM1G (Fig EV5B and Appendix Fig S3K). Concurrently, PPM1G mutants defective in binding to α-catenin, but with intact phosphatase activity (Fig EV5C), were also defective in suppressing cell migration (Fig EV5D and E). Together, these results suggest that PPM1G may act as potential suppressor of tumor cell migration by regulating appropriate localization of α-catenin to membrane and mutations that disrupt its interaction with substrate may contribute to active tumor cell migration.

Analogous to PPM1G, overexpression of either wild-type B56δ or PP2A binding defective mutants (L183A and H282A) suppressed cell migration (Fig 5C and D), whereas Δ91-102 mutant, which is defective in PPM1G binding, was unable to do so. In contrast, loss of B56δ enhanced cell migration (Fig EV5F). Effect of PPM1G and B56δ on cell migration may not be linked to cell proliferation defects as we did not observe any significant changes in rate of proliferation upon either PPM1G or B56δ depletion, within the time frame and conditions utilized during our cell migration assays (Fig EV5G). Further, in support of PPM1G-B56δ complex cooperativity as a holoenzyme, expression of B56δ (Fig EV5H) could not suppress migration in the absence of PPM1G in cells (Fig 5E and F). Likewise, overexpression of PPM1G (Fig EV5I) was unable to hinder migration in B56δ-depleted cells (Fig 5G and H). We further investigated whether the cytoplasmic retention of PPM1G by B56δ is essential and sufficient for controlling cell migration. Wild-type PPM1G was unable to suppress cell migration in cells devoid of B56δ, whereas a constitutive cytoplasmic version of PPM1G (ΔNLS) mutant was sufficient to control cell migration irrespective of B56δ status (Figs EV5J and 5I and J). These data suggest that B56δ-driven cytoplasmic retention of PPM1G in the form of a holoenzyme is critical for controlling cellular migration. Suppression of cell migration by PPM1G-B56δ complex is dependent on dephosphorylation of α-catenin. Expression of a non-phosphorylated form (S641A mutant), but not a constitutive phosphomimetic (S641D) version of α-catenin (Fig EV5K), was able to rescue enhanced migration caused due to either PPM1G depletion or B56δ depletion in cells (Fig 5K and L). Collectively, these results clearly demonstrate that PPM1G-B56δ holoenzyme controls cell migration via dephosphorylation of α-catenin.

Although PPP family of phosphatases are well known to regulate diverse cellular functions via assembling distinct holoenzyme complexes in cell, no PPM phosphatase was shown to function in this manner. Our study has identified one such holoenzyme for PPM1G, which complexes with B56δ regulatory subunit. These findings open up new avenues for PPM family of phosphatases to enhance their substrate repertoire in cells. PPP phosphatases either act a dimer (PP1) or trimeric complex (PP2A) [5]. We found that PPM1G forms a PP1-like dimeric complex and does not require a scaffold subunit. But, there is possibility that other PPM phosphatases can adopt either PP1 dimeric or PP2A-like trimeric holoenzyme assembly. Furthermore, it is also possible that PPM phosphatases can interact with distinct unidentified regulatory proteins, other than classical PPP regulatory subunits. Efforts to identify such holoenzymes in future studies will immensely help us in understanding the substrate and functional diversity for PPM phosphatases.

Adherens junctions are critical for maintenance of structural and functional integrity of multicellular organisms. Loss or abnormal cell–cell adhesion are key hallmarks of malignant transformation,

tumor growth, and metastasis [28]. Previously, it was reported that B56δ null mice develop spontaneous tumors [29] and various B56δ mutations were found to occur in human overgrowth malignancies [30]. Also, increased α-catenin S641 phosphorylation levels were associated with high-grade tumors [24]. Additionally, reports have shown decrease in PPM1G levels in invasive breast and colon cancer [31]. Given that PPM1G and B56δ suppressed cell migration through dephosphorylating α-catenin, it is possible that loss of PPM1G-B56δ holoenzyme leading to increased α-catenin phosphorylation contributes to the invasive capacity of high-grade metastatic tumors. An expression correlation analysis between levels of B56δ, PPM1G, and phosphorylated α-catenin in different tumor grades would further strengthen their physiological role in tumor development and metastasis.

Earlier studies have linked many B56δ missense variations to intellectual disability, autism, macrocephaly, and hypotonia [32–34]. On the other hand, PPM1G knockout mice display severe neural and craniofacial defects leading to embryonic lethality [35]. Interestingly, patients with intellectual disability also show a destabilized cell adhesion complex [36,37]. Thus, it is tempting to speculate that disruption of B56δ-PPM1G-α-catenin-adherens junction axis might be one of the possible mechanisms contributing to development of some of these neurological disorders as well.

# Materials and Methods

### Plasmids and siRNAs

cDNAs of human PP2A regulatory subunits were purchased from Open Biosystems. α-catenin cDNA was a gift from David Rimm (Addgene plasmid # 24194). All cDNAs (PPM1G, α-catenin, B56δ, B56α, B56β, B56γ, and B56ε) were PCR-amplified and cloned into donor vector pDONOR201 (Invitrogen) using gateway cloning and then moved to triple-tagged (S-protein/Flag/streptavidin-binding protein) SFB, Myc, glutathione S-transferase (GST), and MBP-tagged destination vector for expression. All clones were verified by sequencing and checked for expression in 293T cells. Point mutants (PPM1G D496A, D133V, D135N, S173C, G188W, A237D, S240F, A252S, and E258Q) for PPM1G, α-catenin (α-catenin S641A and S641D), PPM1G ΔNLS (deletion of 539-546 residues) mutant, PPM1G deletion mutants, and B56δ deletion mutants were generated by PCR-based site-directed mutagenesis and cloned into SFB, GST, MBP, and Myc-tagged destination vectors. Primers used for cloning various constructs in this study are listed in Table EV2. shRNAs for α-catenin were kindly gifted to us by Dr. Hai-Long Piao and Dr. Li Ma. β-catenin shRNA-1 was a kind gift from Dr. Murali D Bashyam, and shRNA-2 was cloned into pLKO.1 vector. Lentiviral-based shRNAs for PPM1G and B56δ were purchased from Open Biosystems (sequence information along with the catalog numbers included in Table EV2). siRNAs for PPM1G (catalog numbers S102658684 and S102658691) were purchased from Qiagen and transfected using Oligofectamine using standard protocols.

### Antibodies

Following antibodies have been used in this study: PPM1G (1:10,000; Bethyl Laboratories), B56δ (1:10,000; Bethyl

Laboratories), α-catenin (1:5,000; Sigma), phospho-α-catenin (S641) (1:5,000; Sigma), tubulin (1:1,000; Sigma), phospho-α-catenin (S655/T658) (1:5,000; Cell Signaling Technologies), β-catenin (1:10,000; Cell Signaling Technologies), Flag (1:10,000; Santa Cruz Biotechnologies), Myc (1:1,000; Santa Cruz Biotechnologies), actin (1:10,000; Santa Cruz Biotechnologies), E-cadherin (1:5,000; BD Biosciences), FOXK2 (1:5,000; Bethyl Laboratories), GST (1:5,000; Santa Cruz Biotechnologies), HIS (1:1,000; Bethyl Laboratories), MBP (1:5,000; NEB), and histone H3 (1:10,000) (Abcam). Rhodamine-labeled phalloidin to detect F-actin was purchased from Invitrogen. Validation of PPM1G, α-catenin, and β-catenin antibodies for their specificity in immunofluorescence experiments was tested by knocking down each of these genes individually by using specific siRNA/shRNAs (Appendix Fig S4A–C).

## Cell culture and transfection

HEK293T cells were grown in RPMI containing 10% donor bovine serum (DBS) and 1% penicillin and streptomycin. MCF7 and MDA-MB-231 cells were maintained in DMEM plus 10% DBS. MCF10A cells were maintained in DMEM/F12 medium supplemented with 10% DBS, 10 µg/ml insulin, 20 µg/ml EGF, 0.5 µg/ml hydrocortisone, 0.1 µg/ml cholera toxin, and 1% penicillin and streptomycin. Cells were continuously checked by microscopy for their original morphology and tested for mycoplasma contamination by using DAPI staining. Cells were transfected with various plasmids using PEI (Polysciences) or TurboFect (Invitrogen) according to the manufacturer's protocol. Briefly, plasmids (3 µg for a 6-well, 7 µg for a 100-mm dish) were diluted in serum-free media and PEI was added to diluted plasmids in a ratio of 1:3 (DNA amount: PEI volume). The DNA-PEI complexes were incubated for 20 min at room temperature and then added to cells.

## Tandem affinity purification (TAP) and mass spectrometry analysis

HEK293T cells expressing SFB-tagged PP2A regulatory subunits or PPM1G were lysed with NETN buffer (20 mM Tris–HCl, pH 8.0, 100 mM NaCl, 1 mM EDTA, 0.5% Nonidet P-40) containing 1 µg/ml of each pepstatin A and aprotinin on ice for 30 min. Cell debris was removed by centrifugation (15,871 *g*; 10 min; 4°C), and supernatant was incubated with streptavidin Sepharose beads (Amersham Biosciences) for 1 h at 4°C. Then, unbound proteins were removed by washing with NETN three times and bound protein complexes were eluted with 2 mg/ml biotin (Sigma) for 90 min at 4°C. The eluted fraction was incubated with S-protein agarose beads (Novagen) for 1 h at 4°C. The protein complexes bound to S-protein agarose beads were boiled in 2× SDS dye for 5 min and then resolved by SDS–PAGE and stained by Coomassie Blue staining. Entire protein band sample was cut and sent to Taplin Biological Mass Spectrometry Facility at Harvard University for mass spectrometry analysis. Each purification was done in duplicates (*n* = 2).

## Protein analysis by LC–MS/MS

Excised gel bands containing protein complexes were cut into approximately 1-mm³ pieces. Gel pieces were then subjected to a modified in-gel trypsin digestion procedure. Gel pieces were

washed and dehydrated with acetonitrile for 10 min and then completely dried in a speed vac. Gel pieces were rehydrated with 50 mM ammonium bicarbonate solution containing 12.5 ng/µl modified sequencing-grade trypsin (Promega, Madison, WI) at 4°C. After 45 min, excess trypsin solution was removed and 50 mM ammonium bicarbonate solution was added. Samples were incubated at 37°C room overnight. Peptides were extracted with solution containing 50% acetonitrile and 1% formic acid and dried in a speed vac (~1 h). The samples were reconstituted in 5–10 µl of HPLC solvent A (2.5% acetonitrile, 0.1% formic acid). A nanoscale reverse-phase HPLC capillary column was formed by packing 2.6 µm C18 spherical silica beads into a fused silica capillary (100 µm inner diameter x ≈ 30 cm length) with a flame-drawn tip. After column equilibration, sample was loaded through a Famos autosampler (LC Packings, San Francisco CA) onto the column. A gradient was formed, and peptides were eluted with increasing concentrations (gradient of 2–30%) of solvent B (97.5% acetonitrile, 0.1% formic acid) for 1 h. Eluted peptides were subjected to electrospray ionization and then subjected into an LTQ Orbitrap Velos Pro ion-trap mass spectrometer (Thermo Fisher Scientific, Waltham, MA). Peptides were detected, isolated, and fragmented to give a tandem mass spectrum of specific fragment ions for each peptide. Peptide sequences and protein identity were determined by matching protein databases from UniProt (http://www.uniprot.org/taxonomy/9606) with the acquired fragmentation pattern by using Sequest (Thermo Fisher Scientific, Waltham, MA). All databases contained a reversed version of all the sequences, and the data were filtered to 1% peptide false discovery rate.

## Data filtering and data analysis

The interactome data were filtered using CRAPOME tools as described previously [38]. PP2A regulatory subunit interaction of raw data was compared against control GFP, eight other non-phosphatase protein purifications, and CRAPOME control purifications. FCA more than 2 and FCB > 1.5 were used to filter data, and interactions were considered as high confidence interacting partners (HCIPs). Known interactions were identified by comparing with iRef and intact databases.

## RNAi and lentiviral infection

Lentiviral PPM1G, α-catenin, β-catenin, and B56δ shRNA containing plasmids were transfected using PEI (Invitrogen) in BOSC23 packaging cells along with packaging vectors (psPAX2 and pMD2.G). The viral medium was collected 48 and 72 h post-transfection, filtered through 0.45-µm filter (Millipore), and added to the target cells along with polybrene (8 mg/ml). 48 h after transduction, cells were collected and processed for various assays, and immunoblotting was performed with specific antibodies to check the efficiency of knock down. Stable cell lines were generated after selection with 2 µg/µl puromycin.

## Immunoprecipitation and Western blotting

Cells were transfected with various plasmids using polyethyleneimine transfection reagent. For immunoprecipitation assays, cells

were lysed with NETN buffer and cell lysates obtained by centrifugation were incubated with 2 μg of specified antibody bound to either protein A or protein G-Sepharose beads (Amersham Biosciences) or streptavidin Sepharose beads for 1 h at 4°C. The Immunocomplexes were then washed with NETN buffer four times and applied to SDS–PAGE. Immunoblotting was performed following standard protocols.

## Recombinant protein purification

GST and MBP-tagged full-length proteins and mutant plasmids were transformed into *Escherichia coli* BL21 (DE3) cells. Cultures were grown to OD~0.6 and induced with 0.5 mM isopropyl β-D-1-thiogalactopyranoside (IPTG) at 18°C overnight. The cell pellet was lysed in lysis buffer (50 mM Tris [pH 7.5], 150 mM NaCl, and 0.01% NP-40 Igepal and protease inhibitors) and sonicated. Cell lysates were pulled down with either glutathione Sepharose (GST-tagged proteins) or dextran Sepharose beads (MBP-tagged proteins) 2 h at 4°C. Then, beads were washed five times with wash buffer (50 mM Tris [pH 7.5], 300 mM NaCl, 0.01% NP-40 Igepal, 1 mM DTT, and protease inhibitors) and bound proteins were eluted with the elution buffer containing 50 mM Tris (pH 8), 150 mM NaCl, and 10 mM reduced glutathione (for GST-tagged proteins) or 20 mM Tris (pH 7.5), 200 mM NaCl, 1 mM EDTA, and 10 mM maltose (MBP-tagged proteins) for 60 min at 4°C.

## *In vitro* binding assay

Bacterially expressed GST-B56δ or control GST was purified using glutathione Sepharose beads (Amersham) and incubated with bacterial cell lysates expressing MBP PPM1G for 1 h at 4°C. Then, beads were washed three times with wash buffer and bound proteins were eluted by boiling in 2× SDS sample buffer. The proteins were resolved by SDS–PAGE, and the interactions were analyzed by Western blotting.

## Bio-layer interferometry

Bio-layer interferometry (BLI) was done using Octet Red 96 (ForteBio). His-tagged PPM1G and PPP2R1A proteins were immobilized on Ni-NTA biosensors and then incubated with various concentrations of GST-B56δ in kinetics buffer (50 mM Tris [pH 7.5], 300 mM NaCl, 0.01% NP-40 Igepal, 1 mM DTT) at 25 °C. The experiments contained five steps: (i) baseline acquisition; (ii) His-tagged proteins loading onto biosensor; (iii) second baseline acquisition; (iv) association of interacting protein for kon measurement; and (v) dissociation of interacting protein for koff measurement. Data were fitted and analyzed using Octet Data Analysis Software 7.0 (ForteBio).

## *In vitro* phosphatase assay

Bacterially purified substrates were phosphorylated *in vitro* using HEK293T cell lysate in reaction buffer (1 mM ATP, 25 mM HEPES, pH 7.5, 25 mM MgCl$_2$, 2 mM DTT, 25 mM β-glycerophosphate, and 0.1 mM sodium orthovanadate) at 37°C for 90 min. Then, phosphatase reaction was carried out in dephosphorylation buffer (20 mM Tris–HCl pH 7.4, 150 mM NaCl, 5 mM imidazole, 10 mm

MnCl$_2$, and 1 mM DTT) at 30°C for 1 h with 5 μg of bacterially purified recombinant PPM1G proteins. The released phosphate was detected using Malachite Green Assay Kit (Cayman) by measuring the absorbance at 620 nm.

## Immunofluorescence

Cells were cultured on coverslips and fixed with 3% paraformaldehyde solution in PBS containing 50 mM sucrose for 15 min at room temperature. Then, cells were permeabilized with 0.5% Triton X-100 buffer containing 20 mM HEPES pH 7.4, 50 mM NaCl, 3 mM MgCl$_2$, and 300 mM sucrose at room temperature for 5 min and blocked with 1% BSA for 1 h at room temperature. Cells were washed with 1× PBS three times and incubated with various primary antibodies at room temperature for 60 min. After washing with PBS, cells were incubated with FITC or rhodamine-conjugated secondary antibody at room temperature for 1 h. Nuclei were counterstained with 4′,6-diamidino-2-phenylindole (DAPI). Cells were washed with PBS, and coverslips were mounted with glycerin containing paraphenylenediamine and imaged using a confocal microscope (LSM Meta 510, Zeiss). Rhodamine-labeled phalloidin was used for actin cytoskeleton staining. To test the effect of nuclear export on PPM1G localization, cells were processed similarly after they were treated with 20 nM Leptomycin B for 3 h.

The average fluorescence intensity in the nucleus and outside the nucleus (based on DAPI staining) was measured to determine the nuclear-to-cytoplasmic ratio of PPM1G. And the α-catenin levels at the plasma membrane were measured using the quantification method described earlier [39]. Briefly, a straight line was drawn across the cell, and the mean intensity was calculated for a 20-nm area at two independent sites of the cell surface. The mean of these two values was taken as the plasma membrane (PM) fraction, whereas the mean intensity of other areas was taken as the cytoplasmic fraction (Cyt). The ratio of cytoplasm to plasma membrane was calculated and plotted.

## Cell migration assay

The cell migration assay was performed in a 24-well transwell plate with 8-μm polyethylene terephthalate membrane filters (Falcon cell culture insert, BD Biosciences) separating the lower and upper culture chambers. In brief, equal number of cells from different conditions were plated in the upper chamber at 10$^5$ cells per well, in serum-free DMEM. The bottom chamber contained DMEM with 10% FBS. Cells were allowed to migrate for 6 h (MDA-MB-231) and 24 h (MCF10A and MCF7), respectively. After the incubation period, the filter was removed, and non-migrant cells on the upper side of the filter were detached with the use of a cotton swab. Filters were fixed with 4% formaldehyde for 15 min, and cells located in the lower filter were stained with 0.1% crystal violet for 20 min and then imaged using phase-contrast microscopy. Quantification of migrated cells per field was calculated from three independent experiments (derived from average from 3 different fields).

## Statistics

Significance was determined by unpaired Student's *t*-tests for two samples and one-way ANOVAs for multiple samples and further

corrected by Bonferroni's multiple comparison test. Statistical analyses were carried out using GraphPad Prism. Error bars represent standard deviation (SD). The significance levels are indicated by asterisks: $**P < 0.01$ and $***P < 0.001$.

## Data availability

The mass spectrometry proteomic data have been deposited to the ProteomeXchange Consortium via the PRIDE [40] partner repository with the dataset identifier PXD014772.

**Expanded View** for this article is available online.

### Acknowledgements
This work was supported by Wellcome Trust/DBT India Alliance senior fellowship grant (IA/S/16/2/502729 to S.M) and CDFD core funds. P.K and P.T acknowledge the fellowship support from Council of Scientific and Industrial Research (CSIR), and University Grants Commission (UGC), India, respectively. We thank all members of LCDCS for their critical inputs. The authors thank Nanci Rani for providing technical assistance, and sophisticated equipment facility at CDFD for assistance in confocal imaging. We thank Swanth Suresh Kumar for his assistance in shRNA cloning. We thank Dr. Murali D Bashyam for providing β-catenin shRNA, and Dr. Hai-Long Piao and Dr. Li Ma for providing α-catenin shRNAs. We thank Dr. Rakesh Mishra, Director of CSIR-CCMB, for allowing us to utilize CCMB facilities for carrying out biophysical experiments. We thank Dr. Amit Asthana and Sowmya Kranthi from CCMB for their assistance during BLI measurements.

### Author contributions
SM conceptualized and managed the project. SM and PK designed the experiments, analyzed the data, and wrote the manuscript. PK performed most of the experiments; PT performed some of the critical pull-down experiments, and antibody validation along with competition experiments, and NC contributed in purification of PPM1G complex.

### Conflict of interest
The authors declare that they have no conflict of interest.

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
