## [Review Process File · EMBO Reports]

PPM1G forms a PPP-type phosphatase holoenzyme with B56 δ that maintains adherens junction integrity

Parveen Kumar, Prajakta Tathe, Neelam Chaudhary & Subbareddy Maddika

Review timeline:

Submission date:	25 August 2018
Editorial Decision:	4 October 2018
Resubmission:	3 April 2019
Editorial Decision:	21 May 2019
Revision received:	26 July 2019
Accepted:	2 August 2019

Transaction Report:

1st Editorial Decision

4 October 2018

Thank you for the submission of your manuscript to EMBO reports. I apologize for the delay in getting back to you; we have only recently received the final referee report, which I have further discussed with the referees. All reports are copied below.

I am sorry to say that the evaluation of your manuscript is not a positive one. As you will see, while all three referees acknowledge that the findings are potentially interesting, they also raise important concerns regarding data quality, conclusiveness and robustness of the findings.

Due to the nature of the criticisms, the amount of work likely to be required to address them, and the fact that EMBO reports can only invite revision of papers that receive enthusiastic support from the referees upon initial assessment, I am afraid that we do not feel it would be productive to call for a revised version of your manuscript at this stage.

Given the potential interest of your findings, we would, however, have no objections to consider a resubmission of the manuscript in the future if you were able to address all main concerns of the reviewers as highlighted above and in their reports. I would like to stress though that such a manuscript would be treated as a new submission and would be evaluated again, also with respect to the literature and the novelty of your findings at the time of resubmission.

I apologize that I cannot be more positive at this point. I hope, however, that the referee comments are going to be helpful in strengthening your indeed very interesting initial observations and I will be happy to discuss any additional data on this topic with you in the future.

REFeree REPORTS

Referee #1:

The authors propose to have identified a new holo-enzyme PPM1G-B56 δ . Their general message is that the regulatory subunit B56 δ promotes localization of the phosphatase PPM1G to the cytoplasm, where it dephosphorylates α -catenin. They further propose that these events are necessary for assembly of adherent junctions and prevention of cell migration.

Generally, the quality of presented data is not sufficient for publication. Furthermore, the inconsistency of their main statements in different experiments is of great concern, particularly the effect of shRNA B56 δ . Comparing figure 2B and 2K, it appears that the effect of B56 δ knockdown on PPM1G α -catenin interaction is not consistent; also its effect on PPM1G localization is never shown by the authors although B56 δ impact on PPM1G localization is central to their general message.

Finally, we are not convinced that the observed phenotype in figure 4 is due to an effect on cell migration. Indeed, lack of relative quantification of migration values to cell proliferation makes it impossible to conclude whether the results are due to cell proliferation or migration. It seems all the more problematic, as catenins are also implicated in cell proliferation. Furthermore, an effect on cell proliferation would be consistent with the patient and mouse data they cite (spontaneous tumours in B56 δ null mice; high α -catenin S641 phosphorylation levels associated with high grade tumours). Overall lack of consistency in the results presented here questions the reproducibility and significance of the claims.

Referee #2:

In the present study, authors identify PPM1G as an interactor with the PP2A regulatory subunit B56 δ . They go on to show that B56 δ is a regulator of the PPM1G phosphatase activity by altering its localization and this leads to dephosphorylation of α -catenin. This is a potentially interesting discovery as PPM1G is thought to act as a monomer. However, with the small magnitude of change in many of the key experiments and what appears to be a complete lack of any replicates, this manuscript does not provide strong enough evidence for the conclusions that are drawn.

1. The authors need to better define the PPM1G-B56 δ interaction. The B56 family of regulatory subunits have very high protein sequence homology. However, their mass spec data and cytoplasm redistribution indicates the PPM1G interaction was unique to B56 δ . The authors need to verify the specificity of PPM1G-B56 δ interaction by using the other B56 family members as controls for their pulldown assays in figure 1. Mass spec. data also indicate interaction of B56 δ with B56 β and B56 ϵ , has this been examined?

2. The data suggest that B56 δ can only be bound in a complex with PPM1G or PP2A A and C subunits; however, this is a critical aspect to the novelty of the manuscript and needs to be more thoroughly explored. Specifically, if equal amounts of SFB tagged PPM1G and PPP2R1A are mixed with GST- B56 δ , what is the relative binding of the SFB proteins when GST- B56 δ is pulled-down? Additionally, what are the regions of B56 δ and PPM1G that allows their interaction at the exclusion of PP2A A subunit?

3. The in vitro phosphatase assay does not show any sort of specificity for the PPM1G- α -catenin. It is not clear the source of the PPM1G (pulled-down SFB tagged from cells), but another PP2A complex, not containing B56 δ , purified in the same manner should be used as a control for the MBP-tagged α -catenin.

4. The use of mammalian cultured cells is never in vivo, it is in culture or in vitro.

5. The authors use immunofluorescence intensity to determine physiological changes but many of the images have very inconsistent and irregular DAPI stain that makes the staining of other proteins much less convincing. Quantification is required.

6. Figure 1F - From the images, it is difficult to see any cytoplasmic expression of PPM1G. How was the intensity of the cytoplasm/nuclear ratio quantified in Figure 1F? Throughout the paper the authors describe physiological changes associated with using the B56 δ shRNA, but can you see a

decrease in the cytoplasmic localization of PPM1G following B56d shRNA expression by immunofluorescence (Fig. 1F) or the fractionation (Fig. EV1A)?

7. For Figure 2 E-L, the quantitation is included but it is unclear how many times these experiments were performed. In some cases, the changes are only 30% (Fig. 2E, J, K) compared to the control. The authors need to include numbers of replicates and the quantitation across experiments.

8. Figure 3. How many times were these experiments performed and how reproducible are the results? The changes in the localization of the fluorescences should be assessed using some sort of software analysis (ImageJ).

9. Figure 4A-K - All the statistics appear to be technical replicates from one experiment (migrated cells per field). These experiments need to be replicated and comparisons made across experiments.

Referee #3:

Most serine/threonine phosphatase function as holoenzymes in which substrate recognition is mediated by a distinct subunit of the complex. However, thus far PPM phosphatases such as PPM1G have been thought to be an exception and to recognize their substrates without assembling to a holoenzyme complex. The authors report the first holoenzyme of the PPM Ser/Thr phosphatase family. They show that PPM1G binds to PP2A B56d regulatory subunit and identify adherens junction protein α -Catenin as new cytoplasmic substrate of the PPM1G-B56d holoenzyme. This finding is interesting especially from the perspective of existing data on B56d mutations in neural disorders and possible mechanism for disease development involving newly proposed PPM1G-B56d- α -Catenin axis.

General comments:

- Please use the gene nomenclature as PP2A nomenclature with different B-subunits is very confusing. B56g = PPP2R5C
- All IF figures lack specificity controls. Deplete the target proteins by siRNA and show that the signal disappears.
- How does PPM1G-B56d complex assemble? Through B56d sequestering from PP2A? Would it be possible for authors to measure binding affinities for PPM1G-B56d and compare it to similar literature reports for PP2A?

Specific comments that need to be addressed:

Figure 1

B: Replace PPM1G with less exposed image

C: Why GST-B56d and GST alone haven't been used in equimolar amounts? Explain reasoning for stoichiometry used (GST-B56d at present in about 10-fold lower amount than PPM1G) or show titration with different GST-B56d concentrations.

D and E: Show blot for α -Flag in Pulldown samples, not just in the Input. Indicate number of repeats in Figure legend.

F: Are the levels of exogenous B56d physiological? Show blot to demonstrate how do levels of Myc-B56d correlate with the level of endogenous B56d.

Figure EV1

A For cell fractionation experiment, blot both fractions (N and C) α -H3 and tubulin as controls for proper fractioning, and blot all from the same membrane so that proper conclusion can be made on relative N/C distribution of PPM1G.

Show quantifications indicating Mean +S.E.M.

C PPM1G distribution seems to be almost exclusively nuclear and claim that B56d sequesters cytoplasmic PPM1G needs further support.

Figure 2

B, H, L Show less exposed blot for Actin.

B Show less exposed blot for Flag PPM1G.
 Panels E->L indicate in Figure legend how many repeats and show quantification Mean +S.E.M.
 C Show relative expression levels for PPM1G and its D/A mutant.
 G+H Show PPM1G blot.

Figure 3

B Show stainings for a-Catenin and pS641 with the same magnifications

Figure 4

Panels E, G, I, K Show representative Western blots

Figure EV4

B Show less exposed blot for Flag PPM1G, and quantifications with Mean +S.E.M.

Is enhanced cell migration result of only decreased binding of PPM1G to a-Catenin or do PPM1G mutations in the acidic domain affect its catalytic activity for example through allosteric effects?

Pg 11: „PPM1G...regulates proper localisation of a-Catenin to membrane..." Previous data (Fig2 K+L) shows that B56d is needed for localisation and PPM1G for catalytic activity and a-Catenin dephosphorylation- please rewrite to explain the results better.

Materials and methods

Plasmid and siRNAs: reference properly all plasmids used
 List sequences for all primers used in SFB, GST, MBP and Myc-tagged vectors
 Sequence for shRNAs for B56d and PPM1G, and list exact transfection protocol.
 List protocol for LMB treatment.

Minor concerns:

on pg3, change last sentence to „Although many regulatory subunits of PPP phosphatases have been identified, their functions and specific substrate information is still limited."

Figure 3A It appears as though total a-Catenin levels in PPM1G and B56d shRNA would be reduced- comment.

Resubmission

3 April 2019

In reference to our manuscript “ Assembly of PPP-type phosphatase holoenzyme by PPM1G-B56 δ to maintain adherens junction integrity ” with reference number EMBOR-2018-46965V1, I kindly request you to allow us to submit as a revised manuscript.

1. As you may see in the attached revised manuscript and ‘point by point response to the reviewers’, we have addressed all the concerns raised by the reviewers.

2. Importantly, we have extensively characterized PPM1G-B56 δ complex and clearly demonstrated that this is a distinct holoenzyme assembled by PPM phosphatase, independent of PP2A.

3. We have included several sets of new data to clearly support our claims on PPM1G-B56 δ discovery as a new holoenzyme.

Given the potential interest of our findings, as acknowledged by all the three reviewers, with no compromise in the novelty of our identified complex, we believe that our manuscript can be seen as a revised version instead of a new submission. In such a case, may I request you to activate the link from the previously submitted version to enable us to submit a revised manuscript to your office under the same reference number.

I hope you consider our request positively and looking forward to hear from you.

Point by point response to reviewer's comments**Referee #1:**

The authors propose to have identified a new holo-enzyme PPM1G-B56 δ . Their general message is that the regulatory subunit B56 δ promotes localization of the phosphatase PPM1G to the cytoplasm, where it dephosphorylates α -catenin. They further propose that these events are necessary for assembly of adherent junctions and prevention of cell migration. Generally, the quality of presented data is not sufficient for publication. Furthermore, the inconsistency of their main statements in different experiments is of great concern, particularly the effect of shRNA B56 δ .

Response: We considered the concerns raised by the reviewer and accordingly as you may see in the revised manuscript the quality of the data has been improved significantly by repeating several of the critical experiments, particularly in revised figure 1, 2, 3 and also by adding substantial amount of new data. It may be noted that we consistently stated that B56 δ interaction is required for regulating PPM1G localization in cytoplasm (which has been clearly shown in the data from revised figure 2), where B56 δ -PPM1G holoenzyme formation is required for dephosphorylation of α -catenin (revised figure 3).

Comparing figure 2B and 2K, it appears that the effect of B56 δ knockdown on PPM1G α -catenin interaction is not consistent;

Response: The observed differences in the figure 2B and 2K (now revised figures 3B and 3L) was due to differences in the knock down efficiency caused due to shRNAs in different experiments. Now, to support the consistency in our data fully, we repeated these experiments and the quantification data from two independent experiments has been plotted in revised figure EV4.

also its effect on PPM1G localization is never shown by the authors although B56 δ impact on PPM1G localization is central to their general message.

Response: As shown in the revised figure 2E and EV2C, we tested the effect of B56 δ knockdown on PPM1G. By using biochemical fractionation, we clearly demonstrated that depletion of B56 δ in cells prevents cytoplasmic localization of PPM1G. Furthermore, we tested if B56 δ -PPM1G interaction is required for PPM1G presence in the cytoplasm. While expression of full length B56 δ promotes cytoplasmic PPM1G localization, mutant B56 δ that lacks PPM1G binding (Δ 90-102) was unable to retain PPM1G in cytoplasm (Fig 2F & 2G), fully supporting our central message that B56 δ controls PPM1G localization in cells.

Finally, we are not convinced that the observed phenotype in figure 4 is due to an effect on cell migration. Indeed, lack of relative quantification of migration values to cell proliferation makes it impossible to conclude whether the results are due to cell proliferation or migration. It seems all the more problematic, as catenins are also implicated in cell proliferation. Furthermore, an effect on cell proliferation would be consistent with the patient and mouse data they cite (spontaneous tumours in B56 δ null mice; high α -catenin S641 phosphorylation levels associated with high grade tumours). Overall lack of consistency in the results presented here questions the reproducibility and significance of the claims.

Response: As shown in revised figure 5 (figure 4 in previous version), we clearly demonstrated that B56 δ -PPM1G controls cell migration via dephosphorylating α -catenin, through various set of experiments. It may be noted that our phenotype experiments shown here uncouples cell proliferation with cell migration since all our migration assays were done under serum free conditions where the rate of cell proliferation is minimal. Moreover, we always seeded equal number of cells from different conditions for the migration assay that lasts for only 6-10 hours, which may be insufficient to cause alterations in cell proliferation. In addition, effect of PPM1G and B56 δ on cell migration may not be linked to cell proliferation defects as we did not observe any significant changes in rate of proliferation upon either PPM1G or B56 δ depletion, within the time frame and conditions utilized during our cell migration assays (Fig EV6H). Furthermore, previous literature as cited in our discussion, suggested that frequency of B56 δ mutations, lower PPM1G levels and α -Catenin S641 phosphorylation levels were all associated with high-grade tumours, which are metastatic and linked with higher migratory phenotypes rather than cell proliferation. It is possible that development of spontaneous tumors in B56 δ null mice might be due to its

association of PP2A holoenzyme, but its effect on the migration in high grade/metastatic tumors could be explained through PPM1G- α -Catenin axis as shown by us.

Referee #2:

In the present study, authors identify PPM1G as an interactor with the PP2A regulatory subunit B56 δ . They go on to show that B56 δ is a regulator of the PPM1G phosphatase activity by altering its localization and this leads to dephosphorylation of α -catenin. This is a potentially interesting discovery as PPM1G is thought to act as a monomer.

Response: We thank the reviewer for appreciating our study.

However, with the small magnitude of change in many of the key experiments and what appears to be a complete lack of any replicates, this manuscript does not provide strong enough evidence for the conclusions that are drawn.

Response: We added substantial amount of new data along with replicates for several of the key experiments in the revised version of the manuscript to fully support our conclusions.

1. The authors need to better define the PPM1G-B56 δ interaction.

Response: As suggested by the reviewer, we fine mapped the interacting region of B56 δ with PPM1G by generating various deletion mutants. As shown in the revised figures EV1D-EV1G and figures 1F-1H, a region of amino acids between 90-102 of B56 δ is required for its interaction with PPM1G. Interestingly, this region is in the variable N-terminal region of B56 δ , which is distinct from the B56 δ binding region of PP2A (B56 domain) (see revised figures 1G and 1H).

The B56 family of regulatory subunits have very high protein sequence homology. However, their mass spec data and cytoplasm redistribution indicates the PPM1G interaction was unique to B56 δ . The authors need to verify the specificity of PPM1G-B56 δ interaction by using the other B56 family members as controls for their pulldown assays in figure 1.

Response: We agree with reviewer that there is a high sequence similarity between B56 family members. However, it may be noted that these family members show high similarity only in their B56 family conserved domain which is critical for PP2A binding but quite differ in their N and C-terminal regions. In our mapping experiments, we found that PPM1G binds to the N-terminal region of B56 δ , which is unique for each of the B56 member, thus explaining the specificity of B56 δ -PPM1G interaction. In fact, we have checked binding of other B56 family members and found that PPM1G binds specifically to B56 δ not with other family members. (revised Fig. 1E).

Mass spec. data also indicate interaction of B56 δ with B56 β and B56 ϵ , has this been examined?

Response: As seen in our mass spec data, B56 β weakly associates with B56 δ (Fig EV1C) however, PPM1G was found to specifically interact with only B56 δ subunit but not with other members of this family (revised Fig 1E), due to variable N-terminal regions as explained above.

2. The data suggest that B56 δ can only be bound in a complex with PPM1G or PP2A A and C subunits; however, this is a critical aspect to the novelty of the manuscript and needs to be more thoroughly explored. Specifically, if equal amounts of SFB tagged PPM1G and PPP2R1A are mixed with GST- B56 δ , what is the relative binding of the SFB proteins when GST- B56 δ is pulled-down?

Response: As noted by the reviewer, B56 δ assembles two independent holoenzyme complexes either with PPM1G or PP2A as phosphatases. We have substantiated our claims further by demonstrating that B56 δ via two distinct non-overlapping regions interacts with these two complexes independently (as shown in revised figure 1). To measure the relative binding kinetics of B56 δ interaction with two distinct holoenzyme complexes, we utilized bio-layer interferometry. Intriguingly, as shown in revised figure 1I, B56 δ -PPM1G display very close binding kinetics (K_d of 7.1 ± 2.4 nM) as of B56 δ -PPP2R1A interaction ($K_d = 3.07 \pm 1.8$ nM). However, our in vitro competition experiments with excess of recombinant PPM1G did not affect the binding of PPP2R1A with B56 δ (Fig EV1J), therefore suggesting B56 δ assembles distinct PPM1G complex, but not by competing with PP2A holoenzyme complex.

Additionally, what are the regions of B56 δ and PPM1G that allows their interaction at the exclusion of PP2A A subunit?

Response: As described above, based on our deletion and mutation analysis, we found that region spanning 90-102 amino acids of B56 δ binds to PPM1G. This region is exclusive for PPM1G association as deletion of these amino acids did not affect B56 δ interaction with PP2A. On the other hand, B56 δ mutants defective in PPP2R1A binding still binds to PPM1G (revised Fig 1G and Fig 1H).

3. The in vitro phosphatase assay does not show any sort of specificity for the PPM1G- α -catenin. It is not clear the source of the PPM1G (pulled-down SFB tagged from cells), but another PP2A complex, not containing B56 δ , purified in the same manner should be used as a control for the MBP-tagged α -catenin.

Response: We apologize for not clearly mentioning the source of phosphatase in our experimental procedure for phosphatase assay. For all our in vitro phosphatase assays, bacterially purified recombinant PPM1G or its catalytic inactive mutant (D/A) were used. No B56 δ protein is added in the phosphatase assay since B56 δ is required for controlling PPM1G localization but not phosphatase activity.

4. The use of mammalian cultured cells is never in vivo, it is in culture or in vitro.

Response: We replaced the 'in vivo' term with cultured cells in the revised manuscript.

5. The authors use immunofluorescence intensity to determine physiological changes but many of the images have very inconsistent and irregular DAPI stain that makes the staining of other proteins much less convincing. Quantification is required.

Response: We have replaced the IF data by including better quality images in the revised manuscript as suggested by reviewer. Also, we provided the quantification data for all the relevant immunofluorescence images in the revised manuscript (see revised fig 4B and 4G).

6. Figure 1F - From the images, it is difficult to see any cytoplasmic expression of PPM1G. How was the intensity of the cytoplasm/nuclear ratio quantified in Figure 1F? Throughout the paper the authors describe physiological changes associated with using the B56 δ shRNA, but can you see a decrease in the cytoplasmic localization of PPM1G following B56 δ shRNA expression by immunofluorescence (Fig. 1F) or the fractionation (Fig. EV1A)?

Response: It is true that cytoplasmic expression of PPM1G was difficult to capture in cells by using immunofluorescence experiments, possibly due to antibody insensitivity for low amounts of PPM1G. However, cell fractionation experiments followed by western blotting consistently show the presence of significant amount of PPM1G in cytoplasm along with the nuclear fraction. This has been consistent with earlier published data (Liu et al., JBC 2013: 288, 23225-23233), where it has been reported similarly that PPM1G shuttles between cytoplasm and nuclear compartments and dephosphorylate different substrates. As shown in the revised figure 2E and EV2C, we tested the effect of B56 δ knockdown on PPM1G. By using biochemical fractionation, we clearly demonstrated that depletion of B56 δ in cells prevents cytoplasmic localization of PPM1G. Furthermore, we tested if B56 δ -PPM1G interaction is required for PPM1G presence in the cytoplasm. While expression of full length B56 δ promotes cytoplasmic PPM1G localization, mutant B56 δ that lacks PPM1G binding (Δ 90-102) was unable to retain PPM1G in cytoplasm (Fig 2F & 2G), fully supporting our data that B56 δ is required for cytoplasmic PPM1G localization in cells.

7. For Figure 2 E-L, the quantitation is included but it is unclear how many times these experiments were performed. In some cases, the changes are only 30% (Fig. 2E, J, K) compared to the control. The authors need to include numbers of replicates and the quantitation across experiments.

Response: We apologize for not including the details on replicates for the quantification data related to these figures in the original manuscript. All the quantification data is derived from two independent experiments, which is now indicated in each of the revised figure legends. Revised figures including the data from replicates with statistical information is now included as revised figure EV4. The subtle changes observed with α -Catenin phosphorylation in these figures (now revised figure 3) is probably due to presence of very high levels of endogenous PPM1G already in these cells and also due to differences in the knock down efficiency caused due to shRNAs. However, the quantitation data across experiments clearly support the consistent role played by B56 δ -PPM1G in dephosphorylating α -Catenin in cells.

8. Figure 3. How many times were these experiments performed and how reproducible are the

results? The changes in the localization of the fluorescences should be assessed using some sort of software analysis (ImageJ).

Response: As indicated in the revised figure 4 (earlier figure 3), the data shown is a representation of three independent experiments. As suggested by the reviewer, we now have quantified the localization at different sites by measuring fluorescence intensities using ImageJ. Quantification of plasma membrane (PM) to cytoplasmic ratio of α -catenin intensity per each cell was plotted (revised figure 4B and 4G).

9. Figure 4A-K - All the statistics appear to be technical replicates from one experiment (migrated cells per field). These experiments need to be replicated and comparisons made across experiments.

Response: It is not true that the data in figure 4 (now revised figure 5) was derived from technical replicates from one experiment. We apologize for not clearly mentioning these details in the figure legends. In fact, as mentioned in the revised figure legends, quantification data for migrated cells per field (derived from average of 3 different fields) from three independent experiments is shown.

Referee #3:

Most serine/threonine phosphatase function as holoenzymes in which substrate recognition is mediated by a distinct subunit of the complex. However, thus far PPM phosphatases such as PPM1G have been thought to be an exception and to recognize their substrates without assembling to a holoenzyme complex. The authors report the first holoenzyme of the PPM Ser/Thr phosphatase family. They show that PPM1G binds to PP2A B56d regulatory subunit and identify adherens junction protein α -Catenin as new cytoplasmic substrate of the PPM1G-B56d holoenzyme. This finding is interesting especially from the perspective of existing data on B56d mutations in neural disorders and possible mechanism for disease development involving newly proposed PPM1G-B56d- α Catenin axis.

Response: We sincerely thank the reviewer for appreciating our study and highlighting the novelty of the identified complex.

General comments:

Please use the gene nomenclature as PP2A nomenclature with different B-subunits is very confusing. B56g = PPP2R5C

Response: We consistently used B56 δ throughout the manuscript denoting the protein name instead of a gene name to keep in uniformity with the existing literature on these proteins.

- All IF figures lack specificity controls. Deplete the targets proteins by siRNA and show that the signal disappears.

Response: All the antibodies used in our IF experiments are commercial antibodies, which were very well validated in earlier studies and has been extensively utilized across various studies in the field. These antibodies have been very well shown to be specific in the existing literature. Further, as may be seen in revised figure EV5A, PPM1G signal was lost in cells expressing PPM1G siRNA and changes in α -Catenin localization is observed specifically in those cells where PPM1G is lost, supporting the reported specificity for these antibodies.

How does PPM1G-B56d complex assemble? Through B56d sequestering from PP2A? Would it be possible for authors to measure binding affinities for PPM1G-B56d and compare it to similar literature reports for PP2A?

Response: As shown in revised figure 1, we have extensively characterized the assembly of B56 δ -PPM1G complex. Our data clearly demonstrate that B56 δ assemble PPM1G based holoenzyme complex independent of PP2A. Our data does not support the sequestration model since B56 δ interacts with PPM1G and PP2A via two distinct regions (see revised figure 1G and 1H). Further, by using bio-layer interferometry, we measured the binding kinetics of these distinct complexes. Intriguingly, as shown in revised figure 1I, B56 δ -PPM1G display very close binding kinetics (Kd of 7.1 ± 2.4 nM) as of B56 δ -PPP2R1A interaction (Kd = 3.07 ± 1.8 nM). However, our in vitro competition experiments with excess of recombinant PPM1G did not affect the binding of PPP2R1A with B56 δ (Fig EV1J), therefore suggesting B56 δ assembles distinct PPM1G complex, but not by competing with PP2A holoenzyme complex.

Specific comments that need to be addressed:

Figure 1

B: Replace PPM1G with less exposed image

Response: We have replaced the image as suggested.

C: Why GST-B56d and GST alone haven't been used in equimolar amounts? Explain reasoning for stoichiometry used (GST-B56d at present in about 10-fold lower amount than PPM1G) or show titration with different GST-B56d concentrations.

Response: GST and GST-B56d were in fact added at equal concentration in this experiment (revised figure EV1A; earlier figure 1C). However, GST-B56d appears lower in amount in comparison with GST and PPM1G due to protein degradation. The differences in the amount of GST and GST-B56d is due to stability of these proteins on beads. However, it may be noted that even though GST-B56d is relatively low compared to GST alone, PPM1G specifically interacted with GST-B56d but not GST alone. Indeed, this has been observed consistently in our binding experiments with recombinant proteins (also seen in the revised figure 1C).

D and E: Show blot for a-Flag in Pulldown samples, not just in the Input. Indicate number of repeats in Figure legend.

Response: We have repeated this experiment and as shown in revised figure 2D, we included pull down blots along with the input blots.

F: Are the levels of exogenous B56d physiological? Show blot to demonstrate how do levels of Myc-B56d correlate with the level of endogenous B56d.

Response: We have checked the levels of endogenous B56d upon SFB-B56d expression and found that exogenous expression is comparable to endogenous B56d levels (see in revised Fig. 2C)

Figure EV1

A For cell fractionation experiment, blot both fractions (N and C) a-H3 and tubulin as controls for proper fractioning, and blot all from the same membrane so that proper conclusion can be made on relative N/C distribution of PPM1G. B Show quantifications indicating Mean +S.E.M.

Response: We have replaced the data as per reviewer suggestion (Fig. 2D). Also, quantification data is now added (revised figure EV2C).

C PPM1G distribution seems to be almost exclusively nuclear and claim that B56d sequesters cytoplasmic PPM1G needs further support.

Response: It is true that cytoplasmic expression of PPM1G was difficult to capture in cells by using immunofluorescence experiments, possibly due to antibody insensitivity for low amounts of PPM1G. However, cell fractionation experiments consistently show the presence of significant amount of PPM1G in cytoplasm along with the nuclear fraction. Therefore, we utilized fractionation experiments in parallel to study PPM1G in cells. As shown in the revised figure 2E and EV2C, we tested the effect of B56d knockdown on PPM1G. By using biochemical fractionation, we clearly demonstrated that depletion of B56d in cells prevents cytoplasmic localization of PPM1G. Furthermore, we tested if B56d-PPM1G interaction is required for PPM1G presence in the cytoplasm. While expression of full length B56d promotes cytoplasmic PPM1G localization, mutant B56d that lacks PPM1G binding (Δ 90-102) was unable to retain PPM1G in cytoplasm (Fig 2F & 2G), fully supporting our data that B56d is required for cytoplasmic PPM1G localization in cells.

Figure 2

B, H, L Show less exposed blot for Actin.

Response: We have now replaced the actin with less exposed images (revised figure 3B, 3I, 3M).

B Show less exposed blot for Flag PPM1G.

Response: Less exposed blot for flag PPM1G is included (revised figure 3B).

Panels E->L indicate in Figure legend how many repeats and show quantification Mean +S.E.M.

Response: All the quantification data is derived from two independent experiments, which is now indicated in each of the revised figure legends. Revised figures including the data from replicates with statistical information is now included as revised figure EV4.

C Show relative expression levels for PPM1G and its D/A mutant.

Response: We would like to clarify that we used bacterially purified recombinant PPM1G or its catalytic inactive mutant (D/A) proteins but not cell expressed proteins, in our in vitro phosphatase assays (revised figure 3C, earlier figure 2C). Equal amounts of recombinant proteins were added in the assay after measuring the protein concentration.

G+H Show PPM1G blot.

Response: PPM1G blot was included as requested (revised figure 3G, 3I)

Figure 3

B Show stainings for α -Catenin and pS641 with the same magnifications

Response: Images with same magnification were now included in the revised figure 4C.

Figure 4

Panels E, G, I, K show representative Western blots

Response: As requested, representative western blots are now shown for these figures. (Revised Fig. EV6 I, J, K and L)

Figure EV4

B Show less exposed blot for Flag PPM1G, and quantifications with Mean +S.E.M.

Response: Flag PPM1G blot has been replaced with a less exposed image (revised Fig. EV6 B) and the quantification data for α -Catenin binding along with the statistical analysis is presented in revised figure EV6C.

Is enhanced cell migration result of only decreased binding of PPM1G to α -Catenin or do PPM1G mutations in the acidic domain affect its catalytic activity for example through allosteric effects?

Response: We have analysed the catalytic activity of all the PPM1G mutants along with wild type PPM1G by using standard pNPP as a substrate. As shown in revised figure EV6D, no significant changes in phosphatase activity were observed in any of the PPM1G mutants. Thus, the enhanced cell migration might be attributed to loss of their binding to α -Catenin.

Pg 11: „PPM1G...regulates proper localisation of α -Catenin to membrane..." Previous data (Fig2 K+L) shows that B56d is needed for localisation and PPM1G for catalytic activity and α -Catenin dephosphorylation- please rewrite to explain the results better.

Response: Necessary changes have been made in the revised version of the manuscript.

Materials and methods

Plasmid and siRNAs: reference properly all plasmids used
List sequences for all primers used in SFB, GST, MBP and Myc-tagged vectors
Sequence for shRNAs for B56d and PPM1G, and list exact transfection protocol. List protocol for LMB treatment.

Response: All the suggested changes to materials and methods section has been included in the revised manuscript. List of all primers used for cloning and shRNA sequence information has been included as table S3.

Minor concerns:

on pg3, change last sentence to „Although many regulatory subunits of PPP phosphatases have been identified, their functions and specific substrate information is still limited."

Response: The sentence has been revised appropriately as suggested.

Figure 3A It appears as though total α -Catenin levels in PPM1G and B56d shRNA would be reduced- comment.

Response: The appearance of low α -Catenin levels under depleted conditions might be due to redistribution of α -Catenin in the cytoplasm (causing the dilution of the IF signal), which otherwise shows crisp and intense membrane staining in control cells. As seen in western data (figure 3F, 3I, 3J, 3K, 3L and 3M), we did not find any significant changes in total α -Catenin levels upon PPM1G or B56 δ depletion.

2nd Editorial Decision

21 May 2019

Thank you for the submission of your revised manuscript to EMBO reports. Former referee 1 was unfortunately not available anymore, but we have now received the reports from referee 2 and 3 that are copied below.

As you will see, both referees acknowledge that the manuscript has been improved during revision, providing stronger evidence for the proposed PPM1G holoenzyme. However, both referees also point out some remaining concerns that should all be addressed in a final revision. High quality IF data demonstrating antibody specificity must be provided, as requested by referee 3. Please also provide a complete point-by-point response to all reviewer concerns. Depending on the data added your manuscript might be sent back to the referees for a quick check.

From the editorial side, there are also a few things that we need:

- Please provide a complete author checklist, which you can download from our author guidelines (<<http://embor.embopress.org/authorguide>>). Please insert information in the checklist that is also reflected in the manuscript. The completed author checklist will also be part of the RPF.

- Please provide all main and EV figures as separate high-resolution production quality files (.eps, .tif, .jpg)

- Supplementary table S2: please remove the legend from the manuscript file and provide it in the first row of the excel sheet. Since this is a complex table it should be uploaded as Dataset EV1. Please also follow the nomenclature "Dataset EV1" throughout the manuscript text.

- Supplementary tables S1, S3: Please follow the nomenclature Table EV1 and Table EV2 and provide the legend in the first row of the excel file (or in a separate tab).

- EV figures: these will be displayed in the html version of the manuscript in a collapsible format (Expanded View). Unfortunately however, we can only typeset up to 5 EV figures and you currently have six. You have several options. Either you combine some data into one figure so that the final number of EV figures is 5. Or you keep the more important data in the Expanded View and move other data to the Appendix. Alternatively you move all Supplementary figures to the Appendix. The Appendix is a single PDF file called *Appendix*, that contains all figures and their legends. It should start with a title page and a short Table of Content. Appendix figures should be referred to in the main text as: "Appendix Figure S1, Appendix Figure S2" etc. See detailed instructions regarding expanded view here: <<http://embor.embopress.org/authorguide#expandedview>>.

- Primary datasets produced in this study, such as the mass spectrometry proteomics, need to be deposited in an appropriate public database (see <<http://embor.embopress.org/authorguide#dataavailability>>).

The accession numbers and database should be listed in a formal "Data Availability " section (placed after Materials & Method) that follows the model below (see also <<http://msb.embopress.org/authorguide#dataavailability>>).

Data availability

- [data type]: [name of the resource] [accession number/identifier/doi] ([URL or identifiers.org/DATABASE:ACCESSION])

- We perform a routine check on all figures prior to publication and noted the following issues that need your attention:

Fig EV1H and I appear overexposed.

Fig EV6B: alpha-catenin appears overexposed.

Fig EV6K: actin appears overexposed.

Fig EV6L: PPM1G and B56delta appear overexposed.

Please provide scans of these Western blots with as minimal figure processing as possible and also provide the unmodified source data for these panels.

- Our data editors from Wiley have already inspected the Figure legends for completeness and accuracy. Please see their suggested changes in the attached Word file. I have also taken the liberty to make some changes to the Abstract and some suggestions for the title. Could you please review it?

- Moreover, we have noticed several quantification graphs based on duplicate measurements (n = 2)

with given error bars and statistical comparisons, which is not good practice and against journal style. Please display these data as scatter blots without statistical evaluation (Fig. EV2A, EV2C, EV3D, EV4, EV6C).

- Finally, EMBO reports papers are accompanied online by A) a short (1-2 sentences) summary of the findings and their significance, B) 2-3 bullet points highlighting key results and C) a synopsis image that is 550x200-400 pixels large (width x height). You can either show a model or key data in the synopsis image. Please note that the size is rather small and that text needs to be readable at the final size. Please send us this information along with the revised manuscript.

We look forward to seeing a revised version of your manuscript as soon as possible.

REFEREE REPORTS

Referee #2:

The authors did a commendable job adding new data and replicating previous results. The revised manuscript is consequently greatly improved and may eventually be suitable for publication in EMBO Reports. There are, however, some remaining issues mostly pertaining to the new data, that require addressing.

1. Fig. 1G,H. Please show SBP/B56d blots both of input and IP, representative of how many experiments?
2. Fig. 1I, BLI data. Because the L183A and H282A mutants of B56d have not previously been characterized, yet are used extensively in later figures, they need to be characterized by BLI as well. Please indicate the number of experiments performed and the nature of the Kd error range (SD?).
3. Fig. 3E,F authors state that pS641 is "significantly reduced" or "significantly enhanced", but don't show quantification.
4. Fig. 4D, please show a-catenin in the IP
5. The authors' finding that B56d forms distinct complexes with PP2A and PPM1G, although credibly supported by data, is surprising given that the binding regions do not overlap. Perhaps the authors could discuss possible mechanisms by which PP2A and PP2C exclude each other from the B56d complex.

Referee #3:

The authors have very comprehensively revised the paper and provide clearly stronger evidence for their claims. I am satisfied on all the other responses but specificity of IF remains still as a concern. Literature is full of data done with unspecific antibodies and this is a very well recognized problem in biomedical research. Therefore any citations to previous use of the same antibodies in publications is not valid. The data in EV5A is far from convincing, the PPM1G is lost only from 50% of cells. In order to make any conclusions of a -or b-catenin regulation by PPM1G, the authors need to do what was asked: To provide high quality IF data demonstrating loss of IF signals with at least two siRNAs per protein for all proteins for which IF data is presented

1st Revision - authors' response

26 July 2019

Point by point response to reviewer's comments

Referee #2:

The authors did a commendable job adding new data and replicating previous results. The revised manuscript is consequently greatly improved and may eventually be suitable for publication in EMBO Reports.

Response: We thank the reviewer for appreciating the study.

There are, however, some remaining issues mostly pertaining to the new data, that require addressing.

1. Fig. 1G,H. Please show SBP/B56d blots both of input and IP, representative of how many experiments?

Response: We have repeated this experiment and as shown in figure 1G,H in the revised version, we included pull down blots along with the input blots. This data is a representation of three independent experiments.

2. Fig. 1I, BLI data. Because the L183A and H282A mutants of B56d have not previously been characterized, yet are used extensively in later figures, they need to be characterized by BLI as well. Please indicate the number of experiments performed and the nature of the Kd error range (SD?).

Response: As suggested by the reviewer, we extended our BLI measurements using L183A and H282A mutants of B56 δ with PPM1G as well as PPP2R1A. As shown in revised figure EVII, B56 δ L183A and H282A mutants are defective in binding with PPP2R1A however, they show binding affinities quite similar to wild type B56 δ with PPM1G, fully in agreement with our biochemical data for these two distinct complexes.

3. Fig. 3E,F authors state that pS641 is "significantly reduced" or "significantly enhanced", but don't show quantification.

Response: The quantification for these figures was originally included in figure EV4 in the previous version, which is now shown in Appendix figure S3B, C in the revised manuscript.

4. Fig. 4D, please show a-catenin in the IP

Response: We have repeated this experiment by including an additional shRNA for each gene and as shown in figure 4C in the revised version, we included pull down blots along with the input blots.

5. The authors' finding that B56d forms distinct complexes with PP2A and PPM1G, although credibly supported by data, is surprising given that the binding regions do not overlap. Perhaps the authors could discuss possible mechanisms by which PP2A and PP2C exclude each other from the B56d complex.

Response: As discussed extensively in our results, we proposed the assembly of two independent complexes with PPM1G and PP2A by distinct pools of B56 δ in cells. We clearly ruled out the possibility of a competition model for the existence of these two complexes in cells. The exclusion of PP2A from PPM1G-B56 δ complex and vice versa may not possibly come into existence with our model supporting the presence of distinct pools of B56 δ for two different complexes, which may not cross-talk with each other.

Referee #3:

The authors have very comprehensively revised the paper and provide clearly stronger evidence for their claims. I am satisfied on all the other responses

Response: We thank the reviewer for appreciating the study.

But specificity of IF remains still as a concern. Literature is full of data done with unspecific antibodies and this is a very well recognized problem in biomedical research. Therefore any citations to previous use of the same antibodies in publications is not valid. The data in EV5A is far from convincing, the PPM1G is lost only from 50% of cells. In order to make any conclusions of a - or b-catenin regulation by PPM1G, the authors need to do what was asked: To provide high quality IF data demonstrating loss of IF signals with at least two siRNAs per protein for all proteins for which IF data is presented

Response: As suggested by the reviewer, we validated the specificity of PPM1G, α -Catenin and β -catenin antibodies in IF experiments by utilizing two independent siRNA/shRNA for each of these genes. As shown in revised Appendix Figure S4, IF signals for each of the antibody were lost upon knocking down these genes individually.

Corresponding Author Name: Subbareddy Maddika

Journal Submitted to: EMBO reports

Manuscript Number: EMBOR-2018-46965